# Delivery Strategies of siRNA Therapeutics for Hair Loss Therapy

**DOI:** 10.3390/ijms25147612

**Published:** 2024-07-11

**Authors:** Su-Eon Jin, Jong-Hyuk Sung

**Affiliations:** Epi Biotech Co., Ltd., Incheon 21984, Republic of Korea

**Keywords:** siRNA, hair loss, delivery, barrier, design principle, strategic research framework

## Abstract

Therapeutic needs for hair loss are intended to find small interfering ribonucleic acid (siRNA) therapeutics for breakthrough. Since naked siRNA is restricted to meet a druggable target in clinic,, delivery systems are indispensable to overcome intrinsic and pathophysiological barriers, enhancing targetability and persistency to ensure safety, efficacy, and effectiveness. Diverse carriers repurposed from small molecules to siRNA can be systematically or locally employed in hair loss therapy, followed by the adoption of new compositions associated with structural and environmental modification. The siRNA delivery systems have been extensively studied via conjugation or nanoparticle formulation to improve their fate in vitro and in vivo. In this review, we introduce clinically tunable siRNA delivery systems for hair loss based on design principles, after analyzing clinical trials in hair loss and currently approved siRNA therapeutics. We further discuss a strategic research framework for optimized siRNA delivery in hair loss from the scientific perspective of clinical translation.

## 1. Introduction

Hair loss (alopecia), a disease of the hair follicles, displays mild to severe symptoms caused by hormonal imbalance, autoimmune responses or adverse events related to drugs [1,2]. It is classified by the main features of hair follicles, including but not limited to inflammation, trauma, fibrosis or even unknown causes [3,4]. In particular, androgenetic alopecia (AGA) is common in clinics for both males and females and induces a miniaturization of hair follicles. Although topical minoxidil and oral finasteride are Food and Drug Administration (FDA)-approved therapeutics for hair loss, they must be improved to account for individual variability of hair growth cycle regulation and dermal papilla abnormalities in hair follicles [5]. An oral Jak inhibitor of a ritlecitinib capsule (Litfulo^TM^, Pfizer, Inc., Mission, KS, USA) has recently been approved by the FDA for severe alopecia areata (AA) [6,7], but hair loss patients are still suffering from pathophysiology accompanied by psychological pressure, a problem that is attracting extensive public attention.

Small interfering ribonucleic acid (siRNA, 18–23-mers in length) has been developed for the expression control of diseased proteins in hair loss, which is an innovative medicine that seeks to prevent protein overexpression causing unpredicted disease states as one of pioneering gene editing technologies [8,9,10]. The siRNA blocking androgen receptor (AR) expression in AGA has already been distributed even for cosmetic use (e.g., CosumeRNA^TM^, Bioneer Corp., Daejeon, Republic of Korea). Other than AR in AGA, siRNA also has promising potential to directly reduce the disease-inducing proteins mediated by autoimmune responses or other signaling pathways in hair loss [11]. However, delivery systems are inextricably linked to the effectiveness of siRNA drugs as they overcome the limitations of naked siRNA (e.g., large molecular weight, negative charge, and instability leading to functional loss) [12,13].

Strengthening siRNA druggability against pathophysiological barriers, via delivery system technologies, can be one of the critical strategies for siRNA therapeutics to maximize designed activity [14,15]. Delivery systems have been underscored to present targetability and persistency, including lipid-based nanocarriers, polymers, metal/metal oxide nanoparticles, and microneedles [16,17]. They can minimize off-target effects of siRNA drugs systematically and/or locally in diseased states and prolong drug efficacy in the sophisticated environments of hair follicles, a dynamic miniorgan surrounded by local connective tissue [18,19,20]. The delivery systems to hair follicles should enhance the effectiveness protecting normal cells or tissues at the intended sites of the patient’s body following cyclic events even in diseased states.

We introduce clinically impacted siRNA delivery systems for hair loss based on the design principles overcoming intrinsic and pathophysiological barriers beyond siRNA engineering. In this review, current clinical trials for hair loss are described regarding intervention, condition, phase and study design, grounded in an overview of hair loss. Designed delivery systems repurposed for therapeutic siRNA are also summarized from fundamental siRNA characteristics to optimized delivery after analyzing current non-clinical and clinical approaches in siRNA platform development for hair loss. We focus on the formulation of nanocarriers rather than structural modification of siRNA. A strategic research framework for siRNA delivery systems in hair loss is further provided as a path toward enhanced performance.

## 2. Phenotypes of Hair Loss

AGA is the most common non-cicatricial alopecia, caused by excessive androgen (dihydrotestosterone, DHT) responses in both male (vertex and frontotemporal regions) and female (crown and top) patients displaying progressive hair loss in adulthood after puberty [3,21]. AA also occurs in both male and female patients, starting in the scalp and beard regions before 30 years of age, and results in a round area of complete hair loss [3,22]. AA is caused by autoimmune destruction of hair follicles presenting as acute patchy hair loss. Telogen effluvium and anagen effluvium lead to excessive shedding, resulting from a greater proportion of hair follicles entering the telogen stage followed by a stressor event (e.g., drugs, fever, and hair breakage) generally caused by radiotherapy or chemotherapy during the anagen stage [23]. Loose anagen syndrome is related to a shorter anagen stage resulting in hair breakage with short and dull hair caused by the keratin (K6HF) mutation [24]. Trichotillomania is a form of hair loss presenting short and fractured hairs resulting from hair-pulling behavior often associated with psychological disorders such as anxiety and negative moods [25]. Traction alopecia is a transient hair loss, which is common in African-American females affected by prolonged application of tight hairstyles or accessories [26].

Compared with non-cicatricial alopecias, cicatricial alopecias display severe symptoms combined with autoimmune attacks or inflammation, including chronic cutaneous lupus erythematosus, lichen planopilaris, and central centrifugal cicatricial alopecia [3,27]. Chronic cutaneous lupus erythematosus, often observed in females, is a discoid lupus erythematosus (50–85%) presenting scaly and erythematous plaques with the carpet tack sign, potentially increasing in patients who have a familial history of lupus or other autoimmune diseases [28]. Lichen planopilaris, referred to as lichen planus, is the selective destruction of hair follicles caused by chemicals, pressure on the hair, and T lymphocyte-mediated autoimmunity [22]. Genetic susceptibility is also a critical pathomechanism affecting hair loss conditions even in rare cases of central centrifugal cicatricial alopecia (hot comb alopecia, PADI3 gene mutation), which is a scarring hair loss, common in middle-aged African-American females displaying inflammation at the edges of balding lesions [29]. Identifying the main features and causes of hair loss symptoms, suspending causative behaviors and drugs, and trying the best available therapy are primary concerns for the prevention and treatment of hair loss.

## 3. Clinical Trials for Hair Loss

Clinical trials for hair loss have been searched in the clinicaltrials.gov database using hair loss as a keyword (assessed on 26 March 2023). The intervention of siRNA was undetected in the search results of clinical trials for hair loss so far because clinically tunable targets for siRNA are still a challenge in terms of intrinsic and extrinsic barriers to siRNA therapeutics. However, topical formulations or nanoplatforms displaying potential clinical impacts have been introduced to improve the hair loss conditions even in combination therapies. Repurposing is an unprecedented strategy for clinical investigation of siRNA therapeutics for hair loss in the development processes of target identification and delivery system design. In particular, delivery systems from the searched clinical trials in hair loss can be repurposed for siRNA therapeutics after interchangeable modification followed by design principles depending on siRNA characteristics. Design principles and delivery systems overcoming critical barriers will be further discussed in the following sections (see Section 4 and Section 5).

### 3.1. Clinical Trials by Segments

Figure 1 presents the clinical trial information of global registrations, conditions, phases, and study designs of clinical trials for hair loss. Focusing on preventive interventions, vaccines (COVID-19 related), food/nutrition, education/behavior changes, environmental alterations, drugs, and unique formulations were applied for hair loss in clinical trials. On the other hand, therapeutic interventions display not only drugs, medical devices, surgery, and diagnostics to guide therapy, but also complex interventions of drug and drug, drug and food/nutrition, or drug and medical device [30]. Prevention or treatment therapy for hair loss is described in clinical development for detailed disease conditions including alopecia (5.3%), AGA (37.0%, male/female pattern hair loss), AA (30.0%; 0.7% with COVID-19 pneumonia), hair thinning (4.4%), alopecia totalis/universalis (3.7%), telogen effluvium (0.7%), scarring alopecia (4.4%; central centrifugal cicatricial alopecia, hair transplantation in cicatricial alopecia, lichen planopilaris, and lichen planus), cancer-related alopecia (10.5%, chemotherapy-induced alopecia, radiotherapy-induced alopecia (x-ray)), actinic keratosis (2.6%), and others (0.7%, Ludwig type 1/2), with or without acne vulgaris or other skin diseases (atopic dermatitis, vitiligo, psoriasis, mycosis fungoides, urticaria, dermatoses, stretch marks, and hidradenitis suppurativa) in the early-to-late phases as interventional (88.4%) or observational (11.4%) trials, and expanded access (0.2%).

### 3.2. Primary Interventions and Delivery Platforms for Hair Loss

The key interventions of clinical trials are enumerated to confirm hair loss therapy for prevention and treatment, in combination with feasible formulations or nanocarriers (Appendix A). They are classified by small molecules, corticosteroids and others (e.g., finasteride, dutasteride, minoxidil, methylprednisolone, triamcinolone acetonide, hydroxychloroquine, and ALRV5XR (shampoo)) [31,32], Jak inhibitors and antibodies (e.g., ifidancitinib, baricitinib, deuruxolitinib, jaktinib, ritlecitinib, ruxolitinib [33], and tofacitinib), cell therapy or cell therapy-related products (e.g., human autologous hair follicle cells, autologous cultured dermal and epidermal cells, adipose-derived stem cell suspension [34], hair stimulating complexes, GID SVF-2, conditioned media from umbilical cord blood-derived stem cell cultures, lipoaspiration, autologous fat grafts enriched with adipose-derived regenerative cells (ADRCs) [35], platelet-rich plasma (PRP) [36]), and medical devices (e.g., light therapy (Theradome LH80 pro, REVIAN 101), Derma pen (microneedle pen), Dermojet (needle-less syringe), DMEP kit (cryotherapy), MTS-01 (microneedle), scalp cooling (Paxman cooling machine), UV, Venus glow^TM^ (skin renewal machine with serum), and HairDx (genetic testing for baldness)) in terms of pharmacology. Apart from targeting androgens in hair loss to simply remove them, a zinc supplement was also applied with a minoxidil solution (5%) [37].

Key delivery platforms are associated with creams (e.g., aldara (imiquimod)), ointments (e.g., crisaborole, diphenylcyclopropenone (DPCP), LEO 124249)), liniment (e.g., CU-40101), sprays (e.g., CU-40102), gels (e.g., nitric oxide, targretin (bexarotene)), solutions (e.g., latanoprost), matrixes (e.g., platelet-rich fibrin), nanospanlastic dispersions (e.g., sodium valproate), nanofats (grafting), and exosomes.

Among the interventions, ALRV5XR shampoo of dietary supplements has particularly been studied in AGA and/or telogen effluvium (AGA or telogen effluvium, NCT04450602 [31]; AGA, NCT04450589 [38]). It significantly increased hair regrowth without adverse events in males and females. The cell therapy products of human autologous hair follicle cells (NCT01286649), autologous cultured dermal/epidermal cells (NCT01451125), and an adipose-derived stem cell suspension (NCT03388840) have also been the focus of hair loss treatments, even though finasteride, or minoxidil, was extensively tried in the clinical trials compared with light therapy or PRP treatment [32,39]. Innovative nanomedicines or device platforms including nanovesicles (sodium valproate, NCT05017454), exosomes (NCT05658094), and microneedles (NCT00713154, NCT05485571) have also been highlighted in clinical trials [40,41,42] (see Section 5.2.2).

## 4. Small Interfering Ribonucleic Acid (siRNA) as a Therapeutic

### 4.1. RNA Interference (RNAi) and Currently Approved siRNA Therapeutics

RNAi is referred to as a biological process of sequence-specific post-transcriptional gene silencing (PTGS) mediated by small RNAs including small interfering RNA (siRNA), short hairpin RNA (shRNA), and micro-RNA (miRNA) [8,43,44]. The siRNA is a synthetic short double-stranded RNA (21–23 nucleotides long, 13 kDa [15]) used to shut down the target protein expression even transiently. It presents the superior efficacy and targetability to silence one target gene as compared to miRNA which simultaneously compromises the expression of multiple target genes (>100) based on a partial complementation to the 3′-untranslated mRNA region [9,45]. Depending on the performance (i.e., mechanism of action), siRNA and miRNA should be selectively approached for clinical development. Although shRNA, a stem-loop RNA, has similarities to siRNA regarding potency and efficacy to target genes, its function is typically mediated by viral vectors, transported to cytoplasm after synthesis in a cell nucleus, and further processed by an RNA-induced silencing complex (RISC) [45,46]. Safety issues associated with shRNA are raised in therapeutic applications due to the viral vector requirement. Hence, the siRNA successfully knocks down the target gene expression of a disease in a sequence-specific manner, functioning as a feasible gene silencing tool in biotechnological advances and displaying druggability.

In light of currently approved siRNA medicines (e.g., Onpattro^®^, patisiran [47]; Givlaari^®^, givosiran [48]; Oxlumo^®^, lumasiran [49]; and Amvuttra^®^, vutrisiran [50] from Alnylam Pharmaceuticals, Inc. (Cambridge, MA, USA) and Leqvio^®^, inclisiran [51] from Novatis (Basel, Switzerland)), the efficacy and safety of siRNA have been promoted for clinical use following in vitro and in vivo stabilization via delivery systems [8,9]. Onpattro (patisiran), for the treatment of polyneuropathy of hereditary transthyretin-mediated amyloidosis, is the first approved siRNA drug containing lipid nanoparticles comprised of four excipients [DLin-MC3-DMA (ionizable cationic lipid, pKa 6.44), 1,2-distearoyl-sn-glycero-3-phosphocholine (DSPC, amphiphilic phospholipid), cholesterol, and polyethylene glycol (PEG) 2000-C-DMG lipid], administered via intravenous infusion. In addition, N-acetylgalactosamine (GalNAc) conjugation technology to siRNA therapeutics targeted to asialoglycoprotein (ASGPR), which is primarily expressed in hepatocytes, are applied in givlaari (givosiran), oxlumo (lumasiran), amvuttra (vutrisiran), and leqvio (inclisiran) [52,53].

Other than Alnylam Pharmaceuticals, Inc., a pioneering company in siRNA therapeutic development [54], small and medium-sized enterprises have also developed RNAi therapeutics, establishing platforms by partnering with big pharma to conduct clinical trials, including: Arrowhead Pharmaceuticals (Pasadena, CA, USA; TRiM^TM^, targeted RNAi molecule containing targeting ligands, linkers, and structures to enhance stability and pharmacokinetic performance; DPC^TM^, dynamic polyconjugates), Avidity Biosciences (San Diego, CA, USA; AOC^TM^, antibody conjugate to siRNA), Dicerna Pharmaceutical, Inc. (Lexington, MA; GalXC^TM^/GalXC-Plus^TM^, proprietary tetraloop-containing RNAi compounds targeting liver/central nervous system, skeletal muscle, and adipose tissue), Dyne Therapeutics (Waltham, MA, USA; FORCE^TM^, human transferrin receptor 1 (hTfR1) targeted antigen-binding fragment (Fab) with clinically validated linker connected to siRNA for muscle diseases [55]), Exicure, Inc. (Chicago, IL, USA; SNA^TM^, spherical nucleic acid including a benign lipid nanoparticle scaffold with multiple siRNAs), ExonanoRNA (Columbus, OH, USA; ExRNA, engineered exosomes with RNA nanoparticles on the surface [56]), Phio Pharmaceuticals (Marlborough, MA, USA; Intasyl^TM^, a precision siRNA immunotherapy), Silenseed, Ltd. (Jerusalem, Israel; LODER^TM^, local drug EluteR with a miniaturized biodegradable polymeric matrix), and Silence Therapeutics (London, UK; mRNAi GOLD^TM^, liver-targeted siRNA molecules tagging GalNAc) [9,57]. Figure 2 presents the platforms for siRNA delivery that are clinically approved or currently under development.

### 4.2. siRNA Targets for Hair Loss

Hair follicle growth and development are regulated by multiple pathways of survival (e.g., Wnt/β-catenin, Sonic hedgehog (Shh), Notch, bone morphogenetic protein (BMP), etc.) and apoptosis [58,59]. In the anagen stage, new lower hair follicles are formed and regulated by insulin-like growth factor (IGF) 1 and fibroblast growth factor (FGF) 7 from the dermal papilla. FGF 5 and epidermal growth factor (EGF) participate in the cessation of the anagen stage, switching to the catagen stage [60]. In the catagen stage, the hair follicle goes into the controlled processes of regression and/or involution reflecting a programmed cell death in most follicular keratinocytes and some follicular melanocytes [61]. After apoptosis of follicular cells at the end of the catagen stage, the dermal papilla shrinks and moves upwards underneath the bulge for the rest to prepare for transition to the telogen stage. If the dermal papilla travel to the bulge fails, the hair is lost, terminating the hair growth cycle and/or observing hairless gene mutation. After inducing the telogen stage, the hair shaft becomes mature and turns into a club hair, and hair shedding from the hair follicle lasts before restarting the anagen stage in the hair growth cycle [5,62].

Hair loss-associated signaling pathways have been targeted to discover therapeutic siRNA, as scientifically evidenced by multiple markers of hair follicles used to identify their structure and cell composition due to their complexity [63]. Signaling pathways targeting hair loss are consistently connected to the regulation of hair growth cycle control of hair morphogenesis and regeneration. Among the pathways that cause hair loss, the siRNA therapeutics have been studied for targeting C-X-C motif chemokine ligand 12 (CXCL12) [64], Th1 transcription factor (T-box21) [65], CXXC finger protein 5 (CXXC5) [66], egg-laying-defective 9 (EglN) hypoxia inducible factor (HIF)-1 prolyl-hydroxylase (EGLN1 or EGLN3), secreted frizzled-related protein 2 (SFRP2) or serpin family F member 1 (SERPINF1) [67], FGF5 or FGF18 [68], and AR [69,70], with doses on the microgram-to-gram scale based on siRNA efficacy. These are summarized in Table 1, which displays delivery systems and experimental outcomes in non-clinical/clinical models for hair loss.

Zheng et al. (2022) explained that the CXCL12 function inhibited hair growth through CXCR4, which is highly expressed with its receptor, CXCR4 in AGA [64]. CXCL12 was functioning as a hub in differentially expressed genes to inhibit the hair growth via dermal papilla cell regulation for abnormal differentiation and inflammation [71]. It was also overexpressed with MICA by the excessive activity of γδT cells in human skin based on physiological stress, inducing an AA-like autoimmune response in human scalp hair follicles [72,73]. Inhibiting CXCL12 might stimulate hair follicle regeneration. The CXCL12 siRNA triggered telogen-to-anagen transition and increased hair growth in length resulted from hair organ culture. The conditioned medium from dermal fibroblasts applied with CXCL12 siRNA, also induced cell proliferation in dermal fibroblasts and outer root sheath (ORS).

Nakamura et al. (2008) approached T-box21 siRNA after conjugating cationized gelatin and further generating cationized gelatin microspheres for controlled delivery [65]. The T-box21 (Tbx21) gene, referred to as the T-bet gene (T-box transcription factor 21), is responsible for cytokine Th1 production [74]. Inhibiting Tbx21 restored hair shaft elongation in C3H/HeJ mouse, a model of AA, via subcutaneous injection.

Ryu et al. (2023) described CXXC5 siRNA from CXXC5 function, which induced hair loss by prostaglandin D2 (PGD2) related to DHT [66], as a negative regulator in the Wnt/β-catenin pathway [75]. When the CXXC5 siRNA was transfected for 72 h into HaCaT cells, it recovered hair growth suppressed by PGD2. EGLN1 or EGLN3 siRNA was also reported by Liu et al. (2022) [67]. HIF-1α tends to be reduced in an AR-dependent manner since DHT, an AR agonist, promotes the functions to inhibit and degrade HIF-1α [76]. EGLN1 or EGLN3 siRNA fostered dermal papilla cell proliferation for hair follicle growth, a prolonged anagen stage and delayed catagen transition culturing human hair follicles, which was transfected with lipofectamine RNAi MAX reagent. SFRP2 [41] or SERPINF1 (Wnt/β-catenin pathway inhibitor) siRNA also had comparable functions to EGLN1 or EGLN3 siRNA, in hair loss.

Since FGF5 and FGF18 are key factors affecting hair loss in cyclic events of hair follicles, the functions of their siRNAs were investigated in Zhao et al. (2021) [68]. FGF5 siRNA effectively prolonged the anagen stage in C57BL/6 mice of the healthy model after intradermal injection in a cholesterol-conjugated format. Hair loss improvement by FGF5-inhibiting compounds from the monoterpenoid family were also evidenced by anagen promotion, increased hair density, and reduced hair fall [77]. Meanwhile, FGF18 siRNA was topically applied with creams after cholesterol conjugation, which had a comparable hair regrowth potential. FGF18 inhibition accelerated the anagen stage entry of hair follicles, promoting hair regeneration [78]. Its signaling function in hair loss was elucidated by maintaining hair follicle stem cells in a quiescent phase as a downstream modulator of Foxp1 [79]. An FGF18-inhibiting compound (cucurbitacin) was also reported as a hair growth promoter in mice.

Yun et al. (2022) observed androgen receptor siRNA (SAMiRNA) at 0.5 mg/mL three times per week and 5 mg/mL once a week for AGA in a clinical study, resulting in a total hair count increase for 24 weeks [80]. After modifying siRNA to PEG and hydrophobic hydrocarbon conjugates at each end of the unmodified DNA/RNA heteroduplex [99.2 ± 5.1 nm (22 °C, 55% ± 5 humidity) and 105.0 ± 2.5 nm (40 °C, 75% ± 5 humidity)], SAMiRNA was formulated in a hair tonic [ethanol (15%, *v*/*v*), niacinamide (1%, *w*/*v*), betaine (1%, *w*/*v*), biotin (0.02%, *w*/*v*), and buffer in aqueous solution] packaged in a piston-type container equipped with a silicon adaptor for scalp massage.

**Table 1 ijms-25-07612-t001:** Current non-clinical/clinical approaches to siRNA therapeutics for hair loss.

siRNA Target	Nanoplatform for Delivery	siRNA Dose	Outcome	Reference
CXCL12	-	6 μg (subcutaneous injection, every 2 days), C3H/HeN mice; 1 μg, hair organ culture	Triggering telogen-to-anagen transition Increasing hair length in hair organ culture	[64]
T-box21	Cationized gelatin conjugation Cationized gelatin microsphere (controlled delivery)	10 μg (subcutaneous injection), C3H/HeJ mice, alopecia areata model	Restoring hair shaft elongation	[65]
CXXC5	-	10 μM, HaCaT	Recovering hair growth suppressed by PGD2	[66]
EGLN1 or EGLN3 SFRP2 or SERPINF1	Lipofectamine RNAi MAX transfection reagent	0.04 nmol/L (2-day transfection), human hair follicle culture	Promoting dermal papilla proliferation and hair follicle growth with prolonged anagen stage and delayed catagen transition	[67]
FGF5 or FGF18	Cholesterol conjugation Cream	20 μM (50 μL, intradermal injection or topical application), C57BL/6 mice, healthy model	Restoring hair growth	[68]
Androgen receptor (SAMiRNA)	PEG ^1^ and hydrophobic hydrocarbon conjugates at each end of unmodified DNA/RNA heteroduplex [99.2 ± 5.1 nm (22 °C, 55% ± 5 humidity) and 105.0 ± 2.5 nm (40 °C, 75% ± 5 humidity)] Hair tonic [ethanol (15%, *v*/*v*), niacinamide (1% *w*/*v*), betaine (1% *w*/*v*), biotin (0.02% *w*/*v*) and buffer in aqueous solution] (0.5 mg/mL and 5 mg/mL)	Androgenetic alopecia, clinical study; 0.5 mg/mL three times per week: 45, male (test article 8; placebo 6) and female (test article 14; placebo 17); 5 mg/mL once a week: 43, male (test article 9; placebo 10) and female (test article 13; placebo 11)	Increasing total hair counts after administering for 24 weeks	[80]
Androgen receptor (asymmetric siRNA, asiRNA)	Cholesterol conjugation, chemical modification	1.0 μM, human dermal papilla cells; 3 μM and 6 μM, ex vivo human hair follicle culture; 0.125 mg, 0.25 mg, 0.5 mg, and 1.0 mg (intradermal injection), C57BL/6 mice; 0.125 mg, 0.25 mg, and 0.5 mg (intradermal injection), androgenetic alopecia mouse model (dihydrotestosterone daily injection, 25 mg/kg)	Decreasing telogen propagation and increasing the mean hair bulb diameter; Attenuating dihydrotestosterone-mediated increases in interleukin-6, transforming growth factor-β1, and dickkopf-1 levels; No significant toxicity	[70]

^1^ Polyethylene glycol.

AR asymmetric siRNA (asiRNA) after chemical modification or cholesterol conjugation was also reported by Moon et al. (2023) [70]. It had no significant toxicity in human dermal papilla cells, but decreased telogen propagation and increased the mean hair bulb diameter based on attenuating DHT-mediated increases of interleukin-6, transforming growth factor-β1, and dickkopf-1 levels in ex vivo human hair follicles. The AR asiRNA promoted hair growth in C57BL/6 mice after administration via intradermal injection.

Overall, the therapeutic development for hair loss is diversely and successfully challenged using siRNA in a multiangle approach based on the proof-of-concept of diseases, beyond AR targeting due to the off-target effects of DHT blockade [32]. The delivery paradigms of drug modification (e.g., functional group and targeting ligands), environment modification (e.g., formulation), and delivery system introduction (e.g., nanoparticles) for optimization should be evolved and repurposed for clinical translation (see Section 5.2) [17].

## 5. Optimized Delivery for siRNA

### 5.1. Barriers and Strategies for siRNA Delivery

#### 5.1.1. siRNA Designs and Characteristics

siRNA is rationally designed considering the functional parameters of GC content (GC stretch < 10) and base preference, which indicate asymmetrical stability of 5′ and 3′ terminals, suggesting thermodynamic instability of a 5′-antisense (guide) strand in the siRNA duplex for RNA-induced silencing complex (RISC) loading, following the several rules of U-Tei, Reynolds, Amarzguiouri and Tuschl [81,82]. Rationally designed siRNA unexpectedly provided high target silencing efficacy at sub-nanomolar concentration levels in vivo, as compared with conventionally designed siRNA identified and selected by the complementary sequences to target mRNA, consisting of 3′-overhang and 20-base duplexes with 35–75% GC content [83]. To address the pharmaceutical issues enhancing the pharmacokinetic profiles of siRNA and preventing off-target effects and undesirable immune reactions, naked siRNA can be engineered by structural modifications in the phosphonate, ribose, and base for enhanced stability chemistry and directly conjugating target moiety (e.g., GalNAc, Arginyl-glycyl-aspartic acid (RGD), and hyaluronan) [81]. Naked or engineered siRNA should be further formulated and/or administered with delivery systems [8,9].

The size, hydrophilicity, and charge of siRNA (~7–8 nm in length and 2–3 nm in diameter; hydrophilic; negative) can be major obstacles to delivery (Figure 3) [9,57]. Although siRNA is incapable of crossing the cell membrane due to its large size and charge repulsion, it is easily excreted into urine. In addition, endosomal escape of siRNA is essential for cellular trafficking to ensure its efficacy. This is achieved by avoiding its lysosomal degradation based on the proton-sponge effect or the colloid osmotic pressure effect of delivery systems, which results in membrane destabilization or swelling, respectively [84,85]. Since siRNA is administered systemically or locally in clinical developments, bioavailability and biodistribution are critical to achieving a knockdown function in vivo even with rapid clearance [8,9].

Unpredicted immune responses of administered siRNA can be induced as a non-self nucleic acid recognized by innate immune systems [86]. The immunogenicity of siRNA is mediated by interleukins, type 1 interferons (IFN-α/β), and tumor necrosis factor-α stimulating pattern recognition receptors such as toll-like receptors (TLRs) and cytoplasmic receptors. RNAi-induced or exogenous siRNA influences are used to activate protein kinase-R (PKR) and TLRs 3, 7, and 8 [87]. The siRNA-based immune stimulation governing IFN-α release primarily depends on siRNA sequence and structure, which is dictated by putative immunostimulatory motifs of siRNA including poly(U) or GU-rich sequences [88] and blunt-ended siRNA constructs [89]. Therefore, siRNA design of sequence selection, chemical modification, formulation, and particulate delivery system is a critical step to developing siRNA therapeutics. Better siRNA validation matching disease indications should also be employed in mature processes of a new promoted pipeline to select the ideal mRNA target sequences.

#### 5.1.2. Risk Mitigation of siRNA Therapeutics for Hair Loss

Concerning off-target effects, immunogenicity, and toxicity according to siRNA characteristics and hair follicle structure affected by cyclic events, risk mitigation approaches of siRNA therapeutics for hair loss can be categorized by siRNA structure modification, delivery system introduction, safety management, and loss-of-function compensation with clustered regularly interspaced short palindromic repeats (CRISPR) technology [90]. First of all, siRNA structures are modified to enhance the intrinsic stability of siRNA and to reduce the immunogenicity using well-tolerated chemical modifications (2′-O-methyl, 2′-fluoro (F), and phosphorothioate (PS), which is nuclease-resistant and compatible with RNase H activity even associated with cytotoxicity from non-specific protein binding, known as antisense oligonucleotides) [8,14]. The siRNA accessibility to target sequences of mRNA, which is affected by siRNA structure design, is also essential for siRNA potency. For example, reduced siRNA activity from 5′-O-methylation was observed in an antisense strand [91], which was confirmed in the base preferences of siRNA at a specific position for asymmetric thermodynamic instability proceeding RNAi phenomena for RISC loading [86]. The methylation at the 5′-end of the sense (passenger) strand was further applied to select RISC loading properly, while maintaining siRNA potency and improving intrinsic stability of siRNA, irrespective of other structural modifications [86,91].

Delivery systems can be introduced to siRNA, thus improving targetability and persistency in vivo based on the compensation of siRNA characteristics preventing rapid renal excretion, charge repulsion, and nuclease cleavage [92]. Nanoparticle formulation, including cationic components (e.g., cationic lipids or polymers) and lipophilic components (e.g., cholesterol, oil, and fat), and conjugation with targeting moieties are generally used for siRNA delivery to give a matrix effect for persistency and receptor-ligand interaction for targetability, avoiding the hinderance associated with the RNAi processes [86]. In the systemic delivery of siRNA, cyclodextrin (e.g., CALAA-01), stable nucleic acid-lipid particles (e.g., SNALP and TKM-PLK1), lipid nanoparticles (e.g., patisiran), and GalNAc conjugation (e.g., revusiran) were used to enhance in vivo performance [92]. On the other hand, eye drops (e.g., SYL040012), intravitreal injection (e.g., QPI-1007), a nasal spray or nebulizer (e.g., ALN-RSV01), a lipid-based carrier ointment (e.g., TD101), and dissolvable microneedle arrays (e.g., TD101) are offered for local delivery of externally accessible or locally restricted targets for reducing pain during administration. Supplementation for AGA therapy can be applied as a combination for hair loss therapy, while also providing formulation flexibility. It has the potential to address the disease by using vitamins/minerals, nutrafol, viviscal [93], caffeine, rosemary oil, pumpkin seed oil, and saw palmetto [94].

Safety concerns remain related to immunogenicity and off-target/on-target toxicity in a dose-dependent manner delivered to undesired cells, but sophisticatedly designed siRNA delivery systems can be a breakthrough improving therapeutic performance built on targetability and persistency [95]. In the case of patisiran, premedication with a corticosteroid, acetaminophen, and antihistamines are also administered at least 60 min before the filtered patisiran applications to prevent an infusion-related toxicity reaction (e.g., hypersensitivity) [96].

Overall, the safety and efficacy of nanoparticle or nanoparticle-based products are at issue based on their clinical needs, administration route, and physiology because “one size does not fit all” [97]. Although their complex size, structure, and properties are still underestimated in the physiological environment, followed by no ultimate clarification, nanoparticle characterization and usage for siRNA therapeutics in clinics should be guided by “Drug Products, Including Biological Products, That Contain Nanomaterials” [98] or “Considering Whether an FDA-Regulated Product Involves the Application of Nanotechnology” [99], published by the FDA. Characterization of the adequacy and complexity of structure and function, the mechanism of action for biological effects, in vivo release, in vitro–in vivo correlation, stability, nanotechnology maturity, and manufacturing changes is recommended to assess nanomaterials based on their engineered dimensions, structure, and function.

To compensate for loss-of-function mediated by siRNA due to the previously or complementarily expressed disease proteins regulated in multiple signal pathways, CRISPR interference (CRISPRi) of transcription repression can be introduced with siRNA to enhance siRNA potency and efficacy, as distinguished from off-target effects [100,101]. Temporary knockdown effects of siRNA via PTGS are allegedly consolidated by CRISPRi inducing prolonged reversible knockdown via genome editing with low risk of immune stimulation based on CRISPR format regulation [100,102]. In clinical development, CRISPR-Cas9-based ex vivo gene editing has already been introduced, requiring genetic stability in cells [103] including stem cells, induced pluripotent stem cells (iPSCs), and chimeric antigen receptor T (CAR-T) cells. CRISPR/Cas9-based ex vivo cell therapy, exagamglogene autotemcel (Casgevy^TM^, exa-cel; Vertex Pharmaceuticals, Inc., Boston, MA, USA), has also been approved by the FDA [104,105].

### 5.2. siRNA Delivery Principles for Hair Loss

#### 5.2.1. Design Factors of siRNA Delivery Systems

The siRNA delivery systems have been developed based on the design factors involving siRNA itself, the surroundings around siRNA, and delivery systems overcoming challenges of molecular interactions in the body to enhance the delivery potential to target site, minimize the off-target effects, and ultimately improve patient compliance [17] (Figure 4). Design factors of delivery systems have been considered to meet the quality target product profiles (QTPP) of siRNA therapeutics (e.g., intended use, administration route, dosage form, delivery system, dosage strength, container closure, and product quality) provided by the International Council for Harmonisation of Technical Requirements for Pharmaceuticals for Human Use (ICH) Q8 in a document entitled “Pharmaceutical development” [11,106], and to optimize the clinical setting. In siRNA delivery, they lead to a paradigm shift modifying siRNA structures (e.g., functional-group modification, targeting-ligand conjugation, and PEGylation), and environments (e.g., formulation: pH, permeation/penetration enhancer, dispersion enhancer, and clearance inhibitor), and introducing delivery systems (e.g., lipid-based nanoparticles, polymers, microneedle patch, and hydrogel) [107,108]. Multiple design factors of modified siRNA and environments can be introduced to siRNA delivery systems based on delivery principles and clinical strategies in the development framework (see Section 5.2.3).

The siRNA technology innovatively advances the goal of enabling a therapeutic benefit beyond small molecules [109]. The challenging issues developing siRNA therapeutics still include improving in-use and storage stability [110], facilitating desirable pharmacokinetics (e.g., half-life) [111,112], crossing the target cell membrane [113,114], accessing the cytosol for siRNA processing [115,116], reducing immunogenicity [89,117,118], and preventing off-target gene via editing [119,120,121]. In particular, cell internalization and intracellular trafficking for endosomal escape of siRNA are required as a primary fate to be effective, conjugating endolytic peptides or lipids [122,123]. The siRNA processing should be premised on non-toxic and non-immunogenic events. Environment factors can be modified to maximize siRNA efficacy regarding formulation and administration optimized for disease conditions [16,17]. The environmental design factors are generally altered to add immunomodulators and/or endosomal release enhancers. Patisiran was clinically impacted using premedication and ionizable cationic lipid, respectively, as mentioned in Section 4.

The optimized siRNA, followed by modifying siRNA itself and its environment, can be non-clinically or clinically applied with delivery systems based on integrated technology [15,124]. Novel components or repurposed delivery systems can be utilized for siRNA therapeutic development [12,125]. New lipids were synthesized to manufacture siRNA-specific delivery systems [126] (e.g., DLin-MC3-DMA for patisiran (Alnylam), cKK-E12 [127], C12-200 [128], ALC-0315 (Pfizer/BioNTech/Acuitas), SM-102 (Moderna), FTT5 (Ohio State and Beam Therapeutics) [129], and LP01 (Intellia Therapeutics)) [15,130]. Lipid-based nanocarriers are also optimized for hair loss, which have been intensively applied for siRNA delivery, topical/transdermal delivery or hair follicle targeting in non-clinical and clinical development (see Section 5.2.2) [130,131]. Liposomal foam (Sesderma, Miami, FL, USA)/delivery systems (Lipogaine, Washington, DC, USA), niosomal hair masks (Identik Paris, Paris, France), micelle technology shampoo (TRESemme, New York, NY, USA), nanosome delivery systems (DS Laboratories, Miami, FL, USA), and ethosomal formulations (Sinere, Germany) [16] have been marketed as hair loss treatments. Size/polydispersity, morphology, zeta potential, stability, surface to volume ratio, loading efficiency, release profile, and surface decoration for nanocarriers are included in the design factors of delivery systems as critical quality attributes (CQAs) for enhanced/controlled pharmacokinetics based on siRNA release from delivery systems [106].

#### 5.2.2. Delivery Systems for siRNA in Hair Loss

siRNA delivery systems are necessary for the clinical transition of siRNA targeted to hair loss, which have been studied extensively (see Section 4.2). Key issues in siRNA delivery systems are hair follicle structure linked to the hair growth cycle, alopecia induction mechanisms according to the phenotypes, and effective ingredient selection, to optimize the designs for the amelioration of hair loss as a target indication [132,133]. Considering the complicated hair biology in the dermal papilla niche for optimized delivery, hair follicle structure is critical to the search for scientific evidence for ideal delivery to hair follicles in order to overcome barriers, based on repeated hair growth cycles [36,134]. A hair follicle covered by a connective tissue sheath (CTS) is a framed structure of hair shaft (medulla, cortex, and hair cuticle) surrounded by an inner root sheath (IRS) (companion layer, Henle’s layer, Huxley’s layer, and IRS cuticle) and an ORS on the dermal papilla and matrix from the pore of the skin [5,18]. Three stages of anagen (growth), catagen (regression and involution), and telogen (rest) are involved in cyclic events of hair growth to generate the hair follicle structure in terms of hair follicle morphogenesis [135].

Topical delivery or hair follicle targeting systems of small molecules can be promisingly adopted for siRNA in terms of repurposing or repositioning, which involve lipid-based nanocarriers (e.g., liposomes, ethosomes, niosomes, exosomes, lipid nanoparticles [126] and nanostructured lipid carriers [15,131,136]), polymers (e.g., chitosan, dendrimers, and cyclodextrins (CDs) [137,138]), metal/metal oxide nanoparticles (e.g., zinc/zinc oxide nanoparticles [37,139]), and microneedles [42,140]). They make up major efforts toward siRNA delivery associated with large molecular weight, negative charge, and hydrophilicity. Considering hair follicle characteristics of diameter and length depending on the hair growth cycle, the particle size and morphology are primary CQAs in QTPPs describing substance properties, functional traits, and clinical aspects of the final product in the intended patients [106,141]. Table 2 lists the critical design factors of nanocarriers for siRNA delivery. Major nanocarriers are briefly elucidated as follows.

Lipid-based nanocarriers have been developed for hair loss, including liposomes, ethosomes, niosomes, exosomes, lipid nanoparticles, and nanostructured lipid carriers [15,32,36], since lipids are closely connected to hair loss based on lipid homeostasis for normalization of hair growth [62,142]. Lysophosphatidic acid (LPA), an active lipid displaying many biological functions, promotes hair growth in vivo [143] by participating in hair follicle development [144]. In addition, adipocyte interactions in hair growth by platelet derived growth factor (PDGF) may possibly activate the telogen to anagen transition, and mature adipocyte secretes BMP, classic adipokines, and long-chain free fatty acids (FFAs), influencing hair growth [145]. In the context of lipid function for hair growth, vesicular systems of liposomes [146], ethosomes [147,148], and niosomes [149] are widely used as topical delivery systems at nano- to submicron-scales with regular or irregular sphere shapes. The siRNA can be loaded into the aqueous core of the central cavity or membrane in a vesicle functioning as a cargo and interacting with the positively charged vesicle surface via charge–charge interactions [150]. In the case of liposomes, topical application to the skin with the lacZ reporter gene was feasible to deliver the selective gene to the hair follicle [151]. As compared with liposomes of phospholipid bilayers, ethosomes contain ethanol, and stabilizer or edge activator (surfactant) to improve the permeation/penetration function [152,153]. Niosomes comprise non-ionic surfactants, which are generated by self-assembly in aqueous solutions via agitation or temperature elevation [154,155,156]. They advantageously present enhanced chemical stability, have relatively lower production costs than liposomes, which promisingly encapsulated both hydrophobic and hydrophilic drugs for controlled release [157,158]. However, surfactants or charge inducers should be deliberately selected because they possibly cause skin irritation.

Exosomes are naturally occurring extracellular vesicles classified by particle sizes of approximately 50–150 nm produced from cells in lipid-based delivery carriers [41,159,160] prospering as a next-generation delivery platform of drug cargo for hair loss followed by engineering [161,162]. Their characteristics are considerably dependent on the production cell types, stimuli, and treatments due to heterogeneity in size and composition [163,164]. To date, dermal papilla cell- [165] or adipose mesenchymal stromal cell-derived [166] exosomes have been applied for hair loss, displaying hair regrowth based on LEF1 and TGF-β/SMAD3 axis, respectively [167,168].

In nanoparticles of lipid-based nanocarriers, lipid nanoparticles (50–200 nm) [169,170]) and nanostructured lipid carriers (50–500 nm) [171,172,173] are also used as transdermal or topical delivery systems to enhance the particle stability from core lipids, which are advantageous for hair loss [174,175,176,177]. However, inner core structural differences exist in lipid nanoparticles and nanostructured lipid nanocarriers, affecting stability and release depending on the core lipid types. In the core, lipid nanoparticles have a solid lipid, whereas nanostructured lipid carriers contain mixtures of liquid and solid lipids, resulting in highly organized or unorganized lipid cores, respectively [178]. Lipid nanoparticles are further stabilized via PEGylation technology, providing stealth properties according to environmental modulation [179].

In polymer nanoparticles, CDs including α, β, and γ forms are widely adopted natural cyclic oligosaccharides; these are enzymatically degraded from starch, and shaped into truncated cone, bucket, or donut forms, creating an outer surface and an inner cavity [180,181,182]. In particular, β-CD are orally or topically delivered with phytochemicals in AGA patients [183]. Derivatized CDs are functionally modified to obtain the desired characteristics of solubility enhancement and inclusion body generation in pharmaceutical products [184,185,186]). 2-Hydroxypropyl-γ-cyclodextrin (HPγCD) was reported for deep hair follicle targeting, which formed nanoparticles via self-assembly [138,187], although the delivery system for CALAA-01 in clinical trials for cancer already contained β-CD-containing polycations in a formulation with a linear CD-containing polymer, an adamantane-PEG conjugate, and a targeting component of transferrin [188]. The CDs also co-delivered siRNA encapsulated in liposomes [137]. In core–shell structured Fe3O4@ZnO:Er3+,Yb3+@β-CD nanoparticles, drug release after loading was possibly triggered by microwave and Cherenkov radiation via conversion to thermal energy [189].

Zinc/zinc oxide nanoparticles (<100 nm, diverse shapes of rods to spheres) are biocompatible metal/metal oxide nanoparticles [190], potentially associated with hair biology because zinc deficiency causes alopecia in eyelashes and eyebrows [37,191]. Trace quantities of daily zinc at nearly 15 mg are generally required in the diet to maintain homeostasis in the body [192,193] as stated in a clinical trial of chelate zinc supplement (NCT01662089) (see Section 3.2). Zinc oxide and other coordination complexes of zinc (e.g., zinc pyrithione) were utilized as a shampoo to promote hair growth and offer an anti-dandruff function [194,195,196]. Zinc doping in microneedle patches was further applied as a combination therapy for AGA, which contained zinc/copper dual-doped mesoporous silica nanocomposites [139].

Since microneedles (three-dimensional microstructures at micron-scale < 1000 μm [197]) facilitate the macromolecular delivery of hydrophilic drugs, they are considered as alternatives for subcutaneous injection. Microneedles present minimal invasiveness for hair loss treatments in transdermal delivery systems to penetrate the Stratum corneum layer (approximately 10–40 μm thickness [198]) of the skin [42,140]. Microneedle design is beneficial for controlled release into deep locations in the skin, including solid microneedles, coated microneedles, hollow microneedles, dissolvable microneedles, and hydrogel-forming microneedles to avoid first-pass effects and to enhance patient compliance [16,199,200]). Specifically, a lipid–polyvinylamine hybrid nanoparticle-loaded hyaluronic acid dissolving microneedle patch was developed for hair regrowth, which carried miR-218 to activate the Wnt/β-catenin pathway from SFRP2 down-regulation [201,202].

#### 5.2.3. Perspectives on siRNA Delivery Strategies for Hair Loss

Out-of-the-box siRNA delivery systems can be discovered and optimized for translational research in hair loss via leveraging cumulative knowledge based on overcoming challenging barriers [12,130]. Designing engineered delivery systems in terms of major nanoparticle characteristics, including size, shape and surface charge, the nanoparticle interactions with biological environments should be minimized in the journey, which influence cellular uptake, biodistribution, and immune responses from nanoparticle interfaces. Nanoparticle engineering via conjugation of target moiety or incorporation of lipids and/or polymers bearing steric hindrance may avoid interactions and allow them to reach the desired target site [203]. Topical/local delivery systems for hair follicle targeting are also highlighted in hair loss due to easily accessible routes of administration for controlled release, although hair follicle structures and sizes are altered according to cyclic transitions [204]. Moreover, the delivery systems are further optimized for siRNA in alopecia treatment because they are originally matched for small molecules or plasmid DNA proposed for hair loss [205]. Therefore, the strategic framework prefers to build up designing and optimizing siRNA delivery systems in hair loss.

Therapeutic target-based delivery strategies for hair loss are exemplified in a framework successfully dealing with barriers for organs or tissues, cells, and organelles to minimize the off-target losses and to maximize the on-target path [20]. Using a decision matrix, a linear regression, a random forest, a support vector machine, and a neural network in the computational analysis, a rational guided strategy can be a multi-stepwise feedback loop. First of all, it is processed to analyze the biology of target indications in diseases and to set up nanoparticle doses to target an organ. Next, the research data are repeatedly input into a computational algorithm, released from the nanoparticle design, synthesis, and characterization in vitro and in vivo (e.g., cell culture, and animal models). An adjuvant function is eventually used to modulate biological environments at diseased sites or delivery pathways (Figure 5). Small molecules targeting RNA structure can also be added into the research framework with a tight validation of siRNA [206]. The nanoplatform development framework raises issues that must be resolved involving the quality criteria of QTPPs for hair loss therapy, categorized by delivery targets, active agents or cargo, manufacturing process, site of administration, delivery pathway, interaction in the body, and elimination [207] (Table 3).

**Table 2 ijms-25-07612-t002:** Critical design factors of nanocarriers for siRNA delivery.

Nanocarrier	Nanocarrier Description (Strengths and Weaknesses)	Primary Design Factors for siRNA Delivery	References
Liposomes	Closed, spherical phospholipid bilayers; Biocompatible, non-toxic, non-immunogenic, modified with targeting moieties; Low stability, complexity in scale-up manufacturing	Stability; manufacturing	[146]
Ethosomes	Nanovesicles at high ethanol content (20–45%); High elasticity and deformability, enhanced drug permeation and deposition on the skin, possibly added to gels, patches, lotions, etc.; Low drug entrapment, low yield, skin irritation	Stability; toxicity (surfactant); manufacturing	[147,148,152,153]
Niosomes	Nanovesicles containing a nonionic surfactant; Variable and controllable formulations, osmotically active, cost-effective; Time-consuming formulation techniques, incomplete manufacturing process (e.g., hydration), sterilization (e.g., aseptic processing)	Stability (cholesterol); toxicity (surfactant)	[149,154,156,157]
Exosomes	Extracellular vesicles released from cells; High tolerability, cargo-delivery; Heterogeneity, limited reproducibility, low yield, no marketed products	Engineering (cell phenotype; production rate); enrichment; purification; stability; safety	[41,163,164,167,168]
Lipid nanoparticles	Lipid-based nanoparticles consisting of solid lipids or a mixture of solid lipids in the core; High stability, enhanced skin permeation, scale-up manufacturing with sterilization; Low drug accommodation from high crystallinity	Stability; toxicity (surfactant); release (intracellular fate); manufacturing	[126,176]
Nanostructured lipid carriers	Lipid-based nanoparticles containing liquid lipids or a mixture of solid lipids and liquid lipids; High stability; High temperature in hot homogenization (common technique)	Stability; toxicity (surfactant)	[172,174,177,179]
Cyclodextrins	Cyclic oligosaccharides classified as internal cavity (lipophilic) and outer surface (hydrophilic); Masking, protective effect, modified functional group in cyclodextrin structures, controlled release; Difficult release, potential toxicity in structural modification and gelation	Stability; solubility; toxicity; inclusion structure formation	[138,180,183,184,187,208]
Zinc/zinc oxide nanoparticles	Metal/metal oxide nanoparticles; Essential trace element for normal hair growth, improved texture, increased surface area; Safety issues	Stability; dispersity; toxicity; synthesis; purification	[190,191,192,193]
Microneedles (solid, coated, hollow, dissolvable, and hydrogel forming)	Minimally invasive individual needles or a collection of needles at micrometer-scale; Matrix effect, enhanced skin penetration, controlled release; Needle breakage, low mechanical strength	Stability; release; toxicity; manufacturing	[42,140,201]

**Table 3 ijms-25-07612-t003:** Nanoplatform development framework for therapeutic siRNA delivery in hair loss.

Issues	Rationales to Resolve	Quality Criteria in QTPPs ^1^
Delivery target	Hair follicles	Efficacy; potency; (pharmacokinetics/pharmacodynamics)
Active agent or cargo	siRNA or combination; therapeutic moiety release or delivery	Drug release; efficacy; potency
Manufacturing process	Product design; batch scale; identifying critical process parameters and in-process controls (IPCs); consistency and process validation; aseptic processing; good manufacturing practice	Identity; purity/impurities; potency; genetic stability; sterility; IPC release and stability; comparability; bioanalysis; assay development and validation/qualification
Site of administration	Topical or oral; subcutaneous injection	Efficacy; immunogenicity
Delivery pathway	Overcoming barriers into hair follicles via local or systemic route	Efficacy; safety; (pharmacokinetics/pharmacodynamics)
Interaction in the body	Stability enhancement to reach the intended delivery target	Safety; (pharmacokinetics/pharmacodynamics)
Elimination	Toxicity and elimination routes	Safety

^1^ Quality target product profiles.

## 6. Conclusions

siRNA delivery systems have been identified as emerging transformative technologies for a sustainable future. They have been studied in translational research into hair loss to optimize siRNA performance based on required quality criteria. For this purpose, overcoming intrinsic and pathophysiological barriers can be major concerns in siRNA therapeutics to ensure they are safe and effective. In the same context, design principles of siRNA delivery systems are critical for clinical development leading to a delivery paradigm shift from siRNA itself to environments. siRNA delivery systems are integrating the modification of both siRNA and environments, and risks are mitigated to enhance performance at the same time. Strategies to meet the quality criteria of siRNA delivery systems can be built up in a research framework of multi-stepwise feedback loops for design and development supplemented with combinatorial approaches to hair loss therapy.

## Figures and Tables

**Figure 1 ijms-25-07612-f001:**
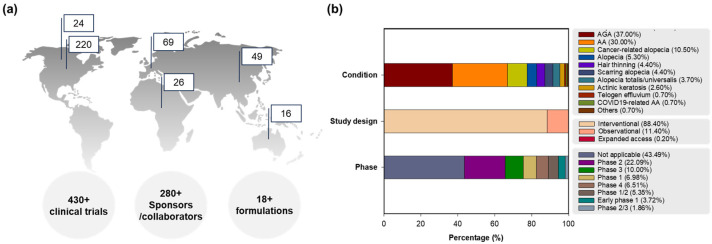
Clinical trial information for hair loss: (**a**) global clinical trial registrations on the map (Top 6 regions) with segmentation by numbers of total clinical trials, sponsors/collaborators, and formulations, (**b**) percentages of clinical trials classified by conditions, phases and study designs. Abbreviations: AA, alopecia areata; AGA, androgenetic alopecia; COVID-19, coronavirus disease 2019.

**Figure 2 ijms-25-07612-f002:**
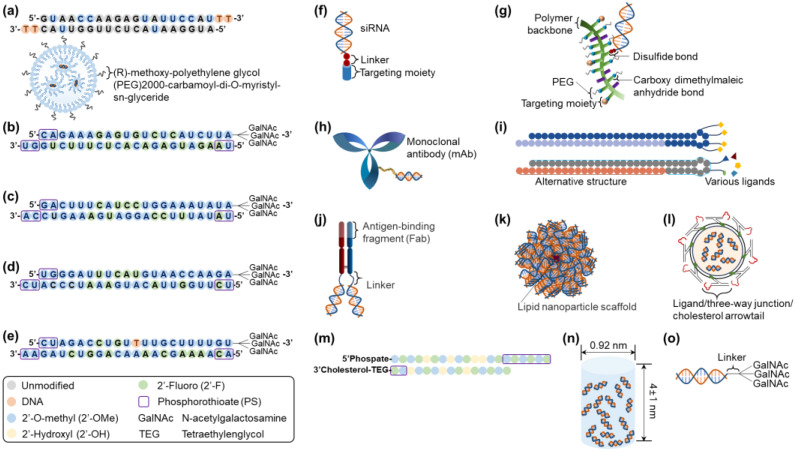
Clinically approved or currently in-development siRNA delivery platforms: (**a**) Onpattro^®^ (patisiran), (**b**) Givlaari^®^ (givosiran), (**c**) Oxlumo^®^ (lumasiran), (**d**) Amvuttra^®^ (vutrisiran), (**e**) Leqvio^®^ (inclisiran), (**f**) TRiM^TM^, (**g**) DPC^TM^, (**h**) AOC^TM^, (**i**) GalXC^TM^/GalXC-Plus^TM^, (**j**) FORCE^TM^, (**k**) SNA^TM^, (**l**) ExRNA, (**m**) Intasyl^TM^, (**n**) LODER^TM^, and (**o**) mRNAi GOLD^TM^. Structural modification patterns for (**a**–**e**) siRNA therapeutics are presented in the box. Abbreviations: siRNA, small interfering ribonucleic acid.

**Figure 3 ijms-25-07612-f003:**
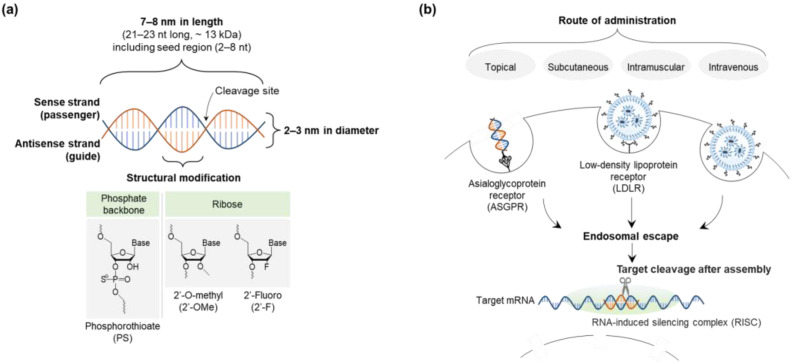
Design and characteristics of siRNA in delivery pathway: (**a**) siRNA structure and size, and (**b**) siRNA delivery pathway. Abbreviations: siRNA, small interfering ribonucleic acid.

**Figure 4 ijms-25-07612-f004:**
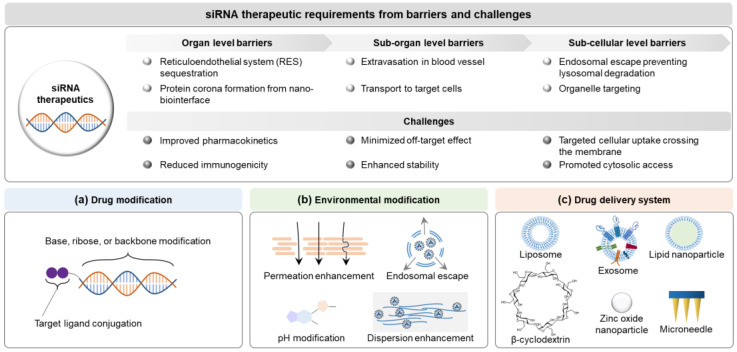
Delivery paradigms for siRNA therapeutics based on the clinical requirements: (**a**) drug modification, (**b**) environment modification, and (**c**) drug delivery system. Abbreviations: siRNA, small interfering ribonucleic acid.

**Figure 5 ijms-25-07612-f005:**
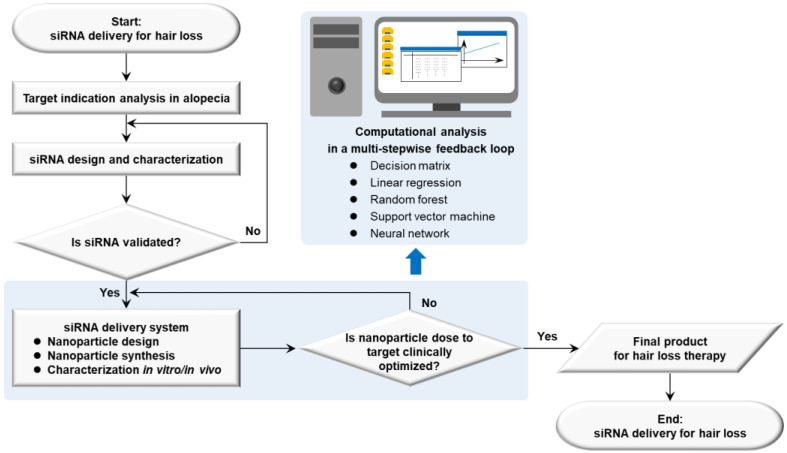
Strategic flow diagram of therapeutic siRNA delivery based on nanoplatform development in hair loss. Abbreviations: siRNA, small interfering ribonucleic acid.

## Data Availability

These data were derived from the following resource available in the public domain: clinicaltrials.gov, summarized in a Appendix A which can be downloaded.

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
