# Peer review of "Delivery Strategies of siRNA Therapeutics for Hair Loss Therapy"

_ijms, 2024, doi:10.3390/ijms25147612_

Round 1

Reviewer 1 Report

Comments and Suggestions for Authors

This is a comprehensive review describing the development of siRNA based therapeutics for hair loss. Overall the manuscript is organized well and easy to follow. However, I have some below suggestions for improvement:

1, the title is not easy to understand. Please revise and make it easy to comprehend. 

2, please include the dose investigated for siRNA therapeutics.

3, please provide a paragraph or table comparing the targets of siRNA for hair loss. 

4, It seems the clinical investigation of siNRA for hair loss is still limited. Please explain why. 

Author Response

Re: Response to Comments and Suggestions for Authors

Manuscript ID: ijms-3015952

Type of manuscript: Review

Title: Delivery strategies of siRNA therapeutics for hair loss overcoming barriers based on clinical impact

June 24, 2024

Dear Editor-in-chief,

                We are pleased to submit our responses to the reviewers’ comments and suggestions. We answered the reviewers’ questions one by one after considerable discussion. The revised manuscript has been corrected and modified according to the reviewers’ comments and the changes in the revised manuscript are presented in red. We hope the revised manuscript to meet your high standards for publication. Thank you for your consideration.

Reviewer #1

This is a comprehensive review describing the development of siRNA-based therapeutics for hair loss. Overall the manuscript is organized well and easy to follow. However, I have some below suggestions for improvement:

  1. The title is not easy to understand. Please revise and make it easy to comprehend.

Thank you for your comment. We modified the title in the revised manuscript as follows: “Delivery strategies of siRNA therapeutics for hair loss therapy”. This manuscript describes siRNA delivery systems for hair loss therapy that overcome intrinsic and extrinsic barriers. We analyzed siRNA delivery systems in previously approved siRNA products and delivery systems for hair loss therapy in clinical trials, which can be repurposed for siRNA therapeutics in hair loss therapy. The siRNA delivery systems developed in current approaches are also discussed based on delivery strategies for siRNA therapeutics in hair loss therapy.

Page on 1: Title

Delivery strategies of siRNA therapeutics for hair loss therapy

Su-Eon Jin* and Jong-Hyuk Sung*

Epi Biotech Co., Ltd., Incheon, Republic of Korea

  1. Please include the dose investigated for siRNA therapeutics.

                Thank you for your comment. The investigated dose of siRNA therapeutics is a key factor in clinical use. In Table 1 of the revised manuscript, siRNA dose levels were additionally added in non-clinical/clinical models as follows. They ranged at 6 μg to 1 mg for topical and intradermal applications. Delivery systems are introduced to prolong the effects of siRNA.

Pages on 30-31: Table 1

Table 1 Current non-clinical/clinical approaches to siRNA therapeutics in hair loss.

Target siRNA

(sequences)

Nanoplatform

for delivery

siRNA dose

Outcome

Reference

CXCL12 siRNA

(5’-GAACAACAACAGACAAGUG-3’,

3’-CUUGUUGUUGUCUGUUCAC-5’)

-

6 μg (subcutaneous injection, every 2 days), C3H/HeN mice;

1 μg, hair organ culture

Triggering telogen-to-anagen transition

Increasing hair length in hair

organ culture

[64]

T-box21 siRNA

(5’-UGAUCGUCCUGCAGUCUCUdTdT-3’,

3’-dTdTACUAGCAGGACGUCAGAGA-5’)

Cationized gelatin conjugation

Cationized gelatin microsphere (controlled delivery)

10 μg (subcutaneous injection),

C3H/HeJ mice, alopecia areata model

Restoring hair shaft elongation

[65]

CXXC5 siRNA

-

10 μM, HaCaT

Recovering hair growth suppressed by PGD2

[66]

EGLN1 or EGLN3 siRNA

SFRP2 or SERPINF1 siRNA

Lipofectamine RNAi MAX transfection reagent

0.04 nmol/L (2-day transfection), human hair follicle culture

Promoting dermal papilla proliferation and hair follicle growth with prolonged anagen stage and delayed catagen transition

[67]

FGF5 or FGF18 siRNA

Cholesterol conjugation

Cream

20 μM (50 μL, intradermal injection or topical application),

C57BL/6 mice, healthy model

Restoring hair growth

[68]

Androgen receptor siRNA

(SAMiRNA)

Polyethylene glycol (PEG) and hydrophobic hydrocarbon conjugates at each end of unmodified DNA/RNA heteroduplex (99.2 ± 5.1 nm (22°C, 55% ± 5 humidity) and 105.0 ± 2.5 nm (40°C, 75% ± 5 humidity))

Hair tonic (ethanol (15%, v/v), niacinamide (1% w/v), betaine (1% w/v), biotin (0.02% w/v) and buffer in aqueous solution)

(0.5 mg/mL and 5 mg/mL)

Androgenetic alopecia, clinical study

0.5 mg/mL three times per week: 45, male (test article 8; placebo 6) and female (test article 14; placebo 17);

5 mg/mL once a week: 43, male (test article 9; placebo 10) and female (test article 13; placebo 11))

Increasing total hair counts after administrating for 24 weeks

[80]

Androgen receptor asymmetric siRNA (asiRNA)

Cholesterol conjugation, chemical modification

1.0 μM, human dermal papilla cells;

3 μM and 6 μM, ex vivo human hair follicle culture;

0.125 mg, 0.25 mg, 0.5 mg, and 1.0 mg (intradermal injection), C57BL/6 mice;

0.125 mg, 0.25 mg, and 0.5 mg (intradermal injection), androgenetic alopecia mouse model (dihydrotestosterone daily injection, 25 mg/kg)

Decreasing telogen propagation and increasing the mean hair bulb diameter; Attenuating dihydrotestosterone-mediated increases in interleukin-6, transforming growth factor-β1, and dickkopf-1 levels; No significant toxicity

[70]

  1. Please provide a paragraph or table comparing the targets of siRNA for hair loss. 

                Although the siRNA targets for hair loss therapy were presented in section “4.2. siRNA targets for hair loss” and Table 1 in the revised manuscript, the targeting approaches were described in greater detail.

Pages on 12: 4.2. siRNA targets for hair loss

~Hair loss-associated signaling pathways have been targeted to discover therapeutic siRNA, as scientifically evidenced by multiple markers of hair follicles used to identify the structure and cell composition due to their complexity [63]. Signaling pathways targeting hair loss are consistently connected to the regulation of hair growth cycle control of hair morphogenesis and regeneration. Among the pathways that cause hair loss, siRNA therapeutics have been studied for targeting C-X-C motif chemokine ligand 12 (CXCL12) [64], Th1 transcription factor  (T-box21) [65], CXXC finger protein 5 (CXXC5) [66], egg-laying-defective 9 (EglN) hypoxia inducible factor (HIF)-1 prolyl-hydroxylase (EGLN1 or EGLN3), secreted frizzled related protein 2 (SFRP2) or serpin family F member 1 (SERPINF1) [67], FGF5 or FGF18 [68], and AR [69, 70], with doses on the microgram-to-gram scale based on siRNA efficacy. These are summarized in Table 1, which displays delivery systems and experimental outcomes in non-clinical/clinical models for hair loss.~

Overall, the therapeutic development for hair loss is diversely and successfully challenged using siRNA in a multiangle approach based on the proof-of-concept of diseases, beyond AR targeting due to the off-target effect of DHT blockade [32]. The delivery paradigms of drug modification (e.g., functional group and targeting ligands), environment modification (e.g., formulation), and delivery system introduction (e.g., nanoparticles) for optimization should be evolved and repurposed for clinical translation (see “5.2. siRNA delivery principles for hair loss”) [17]. ~

Pages on 30-31: Table 1, the same as No.2 answer

  1. It seems the clinical investigation of siRNA for hair loss is still limited. Please explain why. 

                Thank you for your comment. The mechanisms of action for the pathophysiology of hair loss have been investigated, but they are sophisticated and complicated. Although finasteride and minoxidil are available for androgenetic alopecia, and baricitinib was recently approved for alopecia areata, multiple signaling pathways are associated with hair loss therapy in clinics. Clinically tunable targets for siRNA are still a challenge in terms of intrinsic and extrinsic barriers to siRNA therapeutics. Repurposing is an unprecedented strategy for clinical investigation of siRNA therapeutics for hair loss in the development of target identification and delivery system design. We further discuss repurposing from this point of view. The revised manuscript contains this explanation as follows.

Pages on 6-7: 3. Clinical trials for hair loss

Clinical trials for hair loss have been searched in the clinicaltrials.gov database using hair loss as a keyword (assessed on March 26, 2023). The intervention of siRNA was undetected in the search results of clinical trials for hair loss so far because clinically tunable targets for siRNA are still a challenge in terms of intrinsic and extrinsic barriers to siRNA therapeutics. However, topical formulations or nanoplatforms displaying potential clinical impacts have been introduced to improve the hair loss conditions even in a combination therapy. Repurposing is an unprecedented strategy for clinical investigation of siRNA therapeutics for hair loss in the development of target identification and delivery system design. In particular, delivery systems from the searched clinical trials in hair loss can be repurposed for siRNA therapeutics after interchangeable modification followed by design principles depending on siRNA characteristics. Design principles and delivery systems overcoming critical barriers will be further discussed in the following sections (see “4. Small interfering ribonucleic acid (siRNA) as a therapeutic” and “5. siRNA delivery”).

Sincerely,

Su-Eon Jin, Ph.D.

Advisory member, Epi Biotech Co., Ltd.

Jong-Hyuk Sung, Ph.D.

CEO, Epi Biotech Co., Ltd.

Reviewer 2 Report

Comments and Suggestions for Authors

In the manuscript submitted for review, the authors have reviewed the literature on the characterisation of siRNA delivery systems, with a view to their potential use in hair loss therapy. The topic addressed by the authors is important and timely. The manuscript is clearly written and presented in a well-organised manner. The literature cited is mostly recent and relevant publications. Figures and tables are appropriate. Conclusions are consistent with the evidence and arguments presented. Although the article contains enough novelty, I suggest several improvements:

1. In my opinion, the article would have benefited if the authors had formulated the research question and the inclusion and exclusion criteria for publications in the review.

2. Table 1 is too long. I suggest dividing it.

3. Could the authors discuss the current legal status governing the use of nanoparticles including siRNA nanoparticles in therapy?

4. "References" please prepare according to the authors' instructions.

Author Response

Re: Response to Comments and Suggestions for Authors

Manuscript ID: ijms-3015952

Type of manuscript: Review

Title: Delivery strategies of siRNA therapeutics for hair loss overcoming barriers based on clinical impact

June 24, 2024

Dear Editor-in-chief,

                We are pleased to submit our responses to the reviewers’ comments and suggestions. We answered the reviewers’ questions one by one after considerable discussion. The revised manuscript has been corrected and modified according to the reviewers’ comments and the changes in the revised manuscript are presented in red. We hope the revised manuscript to meet your high standards for publication. Thank you for your consideration.

Reviewer #2

In the manuscript submitted for review, the authors have reviewed the literature on the characterisation of siRNA delivery systems, with a view to their potential use in hair loss therapy. The topic addressed by the authors is important and timely. The manuscript is clearly written and presented in a well-organised manner. The literature cited is mostly recent and relevant publications. Figures and tables are appropriate. Conclusions are consistent with the evidence and arguments presented. Although the article contains enough novelty, I suggest several improvements:

  1. In my opinion, the article would have benefited if the authors had formulated the research question and the inclusion and exclusion criteria for publications in the review.

                This manuscript describes siRNA therapeutics for hair loss therapy with repurposed siRNA designs, delivery systems in approved siRNA therapeutics, and clinically tunable formulations from clinical trials for hair loss. We firstly discussed the need for siRNA therapeutics in hair loss therapy and identified barriers to siRNA therapeutics, including siRNA design, currently defined siRNA targets, and delivery systems. Specifically, currently available approaches to target identification and formulation for siRNA were included. However, monogenic conjugation for siRNA delivery, which includes modifying siRNA structure, was excluded from the revised manuscript.

Page on 4: 1. Introduction

~ We introduce clinically impacted siRNA delivery systems for hair loss based on the design principles overcoming intrinsic and pathophysiological barriers beyond siRNA engineering. In this review, current clinical trials for hair loss are described regarding intervention, condition, phase and study design, grounded in an overview of hair loss. Designed delivery systems repurposed for therapeutic siRNA are also summarized from fundamental siRNA characteristics to optimized delivery after analyzing current non-clinical and clinical approaches in siRNA platform development for hair loss. We focus on formulation of nanocarriers rather than structural modification of siRNA. A strategic research framework for siRNA delivery systems in hair loss is further provided as a path toward enhanced performance.

  1. Table 1 is too long. I suggest dividing it.

                Table 1 presents the delivery approaches from clinical trial data for hair loss. We analyzed the clinical trials registered to clinicaltrials.gov, categorized by intervention, dose/delivery system, administration route, and condition. Subcategories of interventions were included as (1) small molecules, corticosteroids and others, (2) Jak inhibitors and antibodies, (3) cell therapy or cell therapy-related products, (4) medical devices, and (5) formulations or delivery systems. Although this table provides useful information, it is too long to put in the revised manuscript. We made it a supplementary table in the revised manuscript according to the reviewer’s suggestion as follows.

Page on 8: 3.2. Primary interventions and delivery platforms for hair loss

The key interventions of clinical trials are enumerated to confirm hair loss therapy for prevention and treatment, in combination with feasible formulations or nanocarriers (Supplementary Table S1). They are classified by small molecules, corticosteroids and others [e.g., finasteride, dutasteride, minoxidil, methylprednisolone, triamcinolone acetonide, hydroxychloroquine, and ALRV5XR (shampoo)] [31, 32], Jak inhibitors and antibodies (e.g., ifidancitinib, baricitinib, deuruxolitinib, jaktinib, ritlecitinib, ruxolitinib [33], and tofacitinib), cell therapy or cell therapy-related products [e.g., human autologous hair follicle cells, autologous cultured dermal and epidermal cells, adipose-derived stem cell suspension [34], hair stimulating complex, GID SVF-2, conditioned media from umbilical cord blood-derived stem cell culture, lipoaspiration, autologous fat graft enriched with adipose-derived regenerative cells (ADRCs) [35], platelet-rich plasma (PRP) [36]], and medical devices [e.g., light therapy (Theradome LH80 pro, REVIAN 101), Derma pen (microneedle pen), Dermojet (needle-less syringe), DMEP kit (cryotherapy), MTS-01 (microneedle), scalp cooling (Paxman cooling machine), UV, Venus glow™ (skin renewal machine with serum), and HairDx (genetic testing for baldness)] in terms of pharmacology. Apart from targeting androgens in hair loss to simply remove, zinc supplement was also applied with a minoxidil solution (5%) [37].

Attachment: Supplementary Table S1

Supplementary Table S1. Delivery approaches based on key interventions in clinical trials for hair loss.

Category

Key intervention

Dose/

Delivery system

Administration route

Condition

NCT number

Small molecules, corticosteroids and others

Finasteride

1 mg

-

Androgenetic alopecia

(female)

NCT01052870

Dutasteride

0.02, 0.1 and 0.5 mg

Oral

Androgenetic alopecia

NCT01231607

0.5 mg

Oral

Male pattern hair loss or androgenetic alopecia

NCT02014584

Minoxidil

5%, q.d. (foam)

2%, b.i.d. (solution)

Topical

Female pattern hair loss

NCT01145625

5% (90% ethanol and 5% propylene glycol)

Topical

Female pattern hair loss

NCT04090801

5% (foam)

Topical

Female pattern hair loss

NCT01226459

5% (noisome, spray)

Topical

Alopecia areata

NCT05587257

Methylprednisolone

(sodium succinate)

15 mg/kg (200 mL fresh orange juice)

Oral

Alopecia totalis

Alopecia universalis

Ophiasic alopecia

NCT01167946

Triamcinolone acetonide

5 mg/mL

Intralesional

Alopecia areata

NCT03535233

Hydroxychloroquine

-

-

Alopecia areata

NCT00176982

ALRV5XR

1 each (capsule)

Oral

Androgenetic alopecia

NCT04450602

3-7 mL (shampoo)

Topical

Telogen effluvium

Hair thinning

Hair loss/baldness

1 each (capsule)

Oral

Androgenetic alopecia

NCT04450589

3-7 mL (shampoo)

Topical

Telogen effluvium

1 mL (serum)

Topical

Hair thinning

Hair loss/baldness

Zinc supplement with minoxidil solution (5%)

15 mg chelate zinc supplement: additional to 5% minoxidil (solution)

Oral

Female pattern hair loss

NCT01662089

Jak inhibitors and antibodies

Ifidancitinib

(ATI-50002)

0.12 and 0.46% (solution)

Topical

Alopecia areata

NCT03354637

Baricitinib

(LY3009104)

2 and 4 mg (tablet)

Oral

Alopecia areata

NCT03899259

2 and 4 mg (tablet)

Oral

Severe or very severe alopecia areata

NCT03570749

High and low (tablet)

Oral

Alopecia areata

Alopecia

Hypotrichosis

Hair diseases

Skin diseases

NCT05723198

Deuruxolitinib

(CTP-543)

8 and 12 mg (tablet)

Oral

Alopecia areata

NCT04797650

Jaktinib

50 and 75 mg (tablet)

Oral

Alopecia areata

NCT05051761

50, 150 and 200 mg (tablet)

Oral

Alopecia areata

NCT04034134

0.5, 1.5 and 2.5% (cream)

Topical

Alopecia areata

NCT04445363

Ritlecitinib

(PF-06651600)

200 mg (q.d., 8 weeks) and then 100 mg (q.d., 40 weeks) (tablet)

Oral

Cicatricial alopecia

NCT05549934

Ruxolitinib

20 mg (tablet)

Oral

Alopecia areata

NCT01950780

0.6% (cream)

Topical

Alopecia areata

NCT02553330

Initial dose and maintenance dose

Oral

Autoimmune polyendocrinopathy candidiasis ectodermal dystrophy (APECED)

Alopecia areata

NCT05398809

Tofacitinib

5 – 10 mg

Oral

Alopecia areata

NCT02299297

Cell therapy or cell therapy-related products

Human autologous hair follicle cells

-

Injection to scalp

Androgenetic alopecia

NCT01286649

Autologous cultured dermal and epidermal cells with 5% minoxidil

-

Injection to scalp

Androgenetic alopecia

Male pattern baldness

Female pattern baldness

NCT01451125

Adipose derived stem cell suspension plus platelet rich plasma

-

Follicular unit extraction

Androgenetic alopecia

NCT03388840

Hair stimulating complex

0.1 mL (30-gauge needle)

Intradermal injection to scalp

Androgenetic alopecia

NCT01501617

Adipose-derived stromal vascular fraction (GID SVF-2)

-

Intradermal injection to scalp

Androgenetic alopecia

NCT02626780

Conditioned media of umbilical cord blood-derived stem cells (NGF-574H)

Hair serum with 5% conditioned media of umbilical cord blood-derived stem cells

Directly used by subjects themselves at home

Androgenetic alopecia

NCT03676400

Biocellular-cellular regenerative mixture

-

Intravenous infusion

Alopecia areata

Scarring alopecia

NCT03078686

Autologous fat graft enriched with adipose-derived regenerative cells (ADRCs)

Purified adipose + 500,000 or + 1,000,000 ADRCs

Subcutaneous injection in scalp

Androgenetic alopecia

NCT02503852

Platelet-rich plasma

6 mL, activated with or without pulsed electrical fields

Subcutaneous injection

Androgenetic alopecia

NCT05348343

-

Injection to half head

Androgenetic alopecia

NCT02074943

-

Injection to scalp

Androgenetic alopecia

NCT03376581

Medical devices

Theradome LH80 pro for photobiomodulation therapy combined with scalp cooling

A wearable laser helmet device, thrice weekly

Light therapy to scalp

Chemotherapy-induced alopecia

NCT05177289

REVIAN 101

A cap for portable use with rechargeable battery and adapter with active LEDs, a daily 10-minute treatment over the course of 26-weeks

Modulated light therapy to scalp

Androgenetic alopecia

NCT04019795

Derma pen

Microneedling combined with methotrexate (25 mg/mL) at 0.02 mL/cm2, a maximum of 0.1-0.2 mL (2.5-5 mg)

Topical

Alopecia areata

NCT05485571

Microneedling after 5% minoxidil application

Topical

Alopecia areata

NCT05587257

Dermojet

A needless syringe, 0.1 mL of Depo-Medrol (methylprednisolone acetate) 40 mg/2 mL

Injection to scalp

Alopecia areata

NCT01017510

Dimethyl ether and propane (DMEP) kit

Superficial cryotherapy using DMEP at -57℃

Topical

Alopecia areata

NCT04680234

MTS-01

7% (gel in 100-mL tube)

Topical

Radiotherapy-induced alopecia

NCT00713154

Paxman cooling machine

-

Topical

Chemotherapy-induced alopecia

NCT01008774

UV

Narrow band UVB (311 nm) phototherapy

Topical

Alopecia areata

NCT03847441

UVB excimer light, twice weekly

Topical

Alopecia areata

NCT01802177

Venus glow™

Venus Glow hydradermabrasion device (cleansing and micromassaging)

Hydradermabrasion on the scalp

Androgenetic alopecia

NCT05426629

HairDx

Sample collection kit for DNA

Saliva collection

Hair loss

Hair loss/baldness

Female pattern baldness

Androgenetic alopecia

NCT04379583

Formulations or delivery systems

Aldara (imiquimod)

5% (cream)

Topical

Alopecia areata

NCT00177021

Crisaborole

Ointment

Topical

Alopecia areata

NCT04299503

Diphenylcyclopropenone (DPCP)

Ointment

Topical

Alopecia areata

NCT03651752

LEO 124249

Ointment

Topical

(eyebrow)

Alopecia areata

NCT03325296

CU-40101

Liniment

Topical

Androgenetic alopecia

NCT05380427

CU-40102

0.25% (2.275mg/mL) (spray)

Topical

Androgenetic alopecia

NCT05135468

XN-001

Nitric oxide gel [14.6 mM sodium nitrite in distilled water with HEC1 (m.w. 50,000-1,250,000);

14.6 mM maleic acid and 14.6 mM ascorbic acid in distilled water with HEC]

Topical

Androgenetic alopecia

NCT01347957

Targretin (bexarotene)

1% (gel)

Topical

Alopecia areata

NCT00063076

Latanoprost

0.005% (solution)

Topical

Alopecia areata

NCT02350023

Ophthalmic solution

Topical (eyelash)

Alopecia areata

NCT00187577

Autologous platelet-rich fibrin matrix (PRFM)

0.1 mL (4-8 mL, isolated from 9-18 mL of peripheral blood)

Intradermal injection to scalp

Alopecia

NCT01590238

Sodium valproate

Nanospanlastic dispersion

Topical

Alopecia areata

NCT05017454

Nanofat

Autologous nanofat grafting

Injection to scalp

Androgenetic alopecia

NCT03506503

Exosomes

100e10 particle (exosomes isolated from human amniotic mesenchymal stem cells)

Injection

Alopecia

Hair loss

(prevention)

NCT05658094

1 Hydroxyethylcellulose

  1. Could the authors discuss the current legal status governing the use of nanoparticles including siRNA nanoparticles in therapy?

                Thank you for your comment. The reference entitled “The regulation of nanomaterials and nanomedicines for clinical application: current and future perspectives” (Rachel Foulkes et al. Biomater. Sci., 2020, 8, 4653-4664. DOI: 10.1039/D0BM00558D.) describes the regulatory needs for nanoparticle products. The safety and efficacy of nanoparticle or nanoparticle-based products are at issue based on their clinical needs, administration route, and physiology because “one size does not fit all”. Nanoparticle usage for siRNA therapeutics in clinics should be guided by “Drug Products, Including Biological Products, that Contain Nanomaterials (FDA, 2022)” or “Considering Whether an FDA-Regulated Product Involves the Application of Nanotechnology (FDA, 2014)” published by the FDA. We additionally added the current legal status for nanoparticles in the revised manuscript as follows.

Page on 20: 5.1.2. Risk mitigation of siRNA therapeutics for hair loss

~ Overall, the safety and efficacy of nanoparticle or nanoparticle-based products are at issue based on their clinical needs, administration route, and physiology because “one size does not fit all” [97]. Although their complexed size, structure, and properties are still underestimated in the physiological environment followed by no ultimate clarification, nanoparticle characterization and usage for siRNA therapeutics in clinics should be guided by “Drug Products, Including Biological Products, that Contain Nanomaterials” [98] or “Considering Whether an FDA-Regulated Product Involves the Application of Nanotechnology” [99], published by the FDA. Characterization of the adequacy and complexity of structure and function, the mechanism of action for biological effects, in vivo release, in vitro-in vivo correlation, stability, nanotechnology maturity, and manufacturing changes is recommended to assess nanomaterials based on their engineered dimension, structure and function. ~

References

  1. Moon, I. J.; Yoon, H. K.; Kim, D.; Choi, M. E.; Han, S. H.; Park, J. H.; Hong, S. W.; Cho, H.; Lee, D. K.; Won, C. H., Efficacy of Asymmetric siRNA Targeting Androgen Receptors for the Treatment of Androgenetic Alopecia. Mol Pharm 2023, 20(1), 128-135.
  2. FDA, Drug Products, Including Biological Products, that Contain Nanomaterials. 2022.
  3. FDA, Considering Whether an FDA-Regulated Product Involves the Application of Nanotechnology. 2014.

  1. "References" please prepare according to the authors' instructions.

                We double-checked the cited references according to the authors’ instruction. In the revised manuscript, the references are as follows.

Pages on 30-51: References

  1. Premanand, A.; Reena Rajkumari, B., Androgen modulation of Wnt/β-catenin signaling in androgenetic alopecia. Arch Dermatol Res 2018, 310(5), 391-399.
  2. Workman, K.; Piliang, M., Approach to the patient with hair loss. J Am Acad Dermatol 2023, 89(2), S3-S8.
  3. Qi, J.; Garza, L. A., An overview of alopecias. Cold Spring Harb Perspect Med 2014, 4(3), a013615.
  4. Wall, D.; Meah, N.; Fagan, N.; York, K.; Sinclair, R., Advances in hair growth. Fac Rev 2022, 11, 1.
  5. Paus, R.; Cotsarelis, G., The Biology of Hair Follicles. N Engl J Med 1999, 341(7), 491-497.
  6. King, B.; Zhang, X.; Harcha, W. G.; Szepietowski, J. C.; Shapiro, J.; Lynde, C.; Mesinkovska, N. A.; Zwillich, S. H.; Napatalung, L.; Wajsbrot, D.; Fayyad, R.; Freyman, A.; Mitra, D.; Purohit, V.; Sinclair, R.; Wolk, R., Efficacy and safety of ritlecitinib in adults and adolescents with alopecia areata: a randomised, double-blind, multicentre, phase 2b-3 trial. Lancet 2023, 401(10387), 1518-1529.
  7. Harrison, C., Hair loss treatments take aim at the immune system. Nat Biotechnol 2023, 41(9), 1179-1181.
  8. Setten, R. L.; Rossi, J. J.; Han, S.-p., The current state and future directions of RNAi-based therapeutics. Nat Rev Drug Discov 2019, 18(6), 421-446.
  9. Hu, B.; Zhong, L.; Weng, Y.; Peng, L.; Huang, Y.; Zhao, Y.; Liang, X.-J., Therapeutic siRNA: state of the art. Signal Transduct Target Ther 2020, 5(1), 101.
  10. Alshaer, W.; Zureigat, H.; Al Karaki, A.; Al-Kadash, A.; Gharaibeh, L.; Hatmal, M. m. M.; Aljabali, A. A. A.; Awidi, A., siRNA: Mechanism of action, challenges, and therapeutic approaches. Eur J Pharmacol 2021, 905, 174178.
  11. Paus, R., Therapeutic strategies for treating hair loss. Drug Discov Today Ther Strateg 2006, 3(1), 101-110.
  12. Sajid, M. I.; Moazzam, M.; Kato, S.; Yeseom Cho, K.; Tiwari, R. K., Overcoming Barriers for siRNA Therapeutics: From Bench to Bedside. Pharmaceuticals 2020, 13(10), 294.
  13. Dowdy, S. F., Overcoming cellular barriers for RNA therapeutics. Nat Biotechnol 2017, 35(3), 222-229.
  14. Kulkarni, J. A.; Witzigmann, D.; Thomson, S. B.; Chen, S.; Leavitt, B. R.; Cullis, P. R.; van der Meel, R., The current landscape of nucleic acid therapeutics. Nat Nanotechnol 2021, 16(6), 630-643.
  15. Paunovska, K.; Loughrey, D.; Dahlman, J. E., Drug delivery systems for RNA therapeutics. Nat Rev Genet 2022, 23(5), 265-280.
  16. Salim, S.; Kamalasanan, K., Controlled drug delivery for alopecia: A review. J Control Release 2020, 325, 84-99.
  17. Vargason, A. M.; Anselmo, A. C.; Mitragotri, S., The evolution of commercial drug delivery technologies. Nat Biomed Eng 2021, 5(9), 951-967.
  18. Schneider, M. R.; Schmidt-Ullrich, R.; Paus, R., The Hair Follicle as a Dynamic Miniorgan. Curr Biol 2009, 19(3), R132-R142.
  19. Costa, C.; Cavaco-Paulo, A.; Matamá, T., Mapping hair follicle-targeted delivery by particle systems: What has science accomplished so far? Int J Pharm 2021, 610, 121273.
  20. Poon, W.; Kingston, B. R.; Ouyang, B.; Ngo, W.; Chan, W. C. W., A framework for designing delivery systems. Nat Nanotechnol 2020, 15(10), 819-829.
  21. Ho, C. H., Sood, T., Zito, P.M., Androgenetic Alopecia. In StatPearls [Internet], Treasure Island (FL): StatPearls Publishing: 2023.
  22. Cardoso, C. O.; Tolentino, S.; Gratieri, T.; Cunha-Filho, M.; Lopez, R. F. V.; Gelfuso, G. M., Topical Treatment for Scarring and Non-Scarring Alopecia: An Overview of the Current Evidence. Clin Cosmet Investig Dermatol 2021, 14, 485-499.
  23. Alhanshali, L.; Buontempo, M.; Shapiro, J.; Lo Sicco, K., Medication-induced hair loss: An update. J Am Acad Dermatol 2023, 89(2), S20-S28.
  24. Dhurat, R. P.; Deshpande, D. J., Loose anagen hair syndrome. Int J Trichology 2010, 2(2), 96-100.
  25. Pereyra, A. D., Saadabadi, A., Trichotillomania. In StatPearls [Internet], Treasure Island (FL): StatPearls Publishing: 2023.
  26. Pulickal, J. K., Kaliyadan, F., Traction Alopecia. In StatPearls [Internet], Treasure Island (FL): StatPearls Publishing: 2023.
  27. Filbrandt, R.; Rufaut, N.; Jones, L.; Sinclair, R., Primary cicatricial alopecia: diagnosis and treatment. Cmaj 2013, 185(18), 1579-85.
  28. Eastham, A. B.; Vleugels, R. A., Cutaneous Lupus Erythematosus. JAMA Dermatology 2014, 150(3), 344-344.
  29. Uitto, J., Genetic Susceptibility to Alopecia. N Engl J Med 2019, 380(9), 873-876.
  30. Craig, P.; Dieppe, P.; Macintyre, S.; Michie, S.; Nazareth, I.; Petticrew, M., Developing and evaluating complex interventions: the new Medical Research Council guidance. BMJ 2008, 337, a1655.
  31. Feldman, P. R.; Fiebig, K. M.; Piwko, C.; Mints, B. M.; Brown, D.; Cahan, D. J.; Guevara-Aguirre, J., Safety and efficacy of ALRV5XR in women with androgenetic alopecia or telogen effluvium: A randomised, double-blinded, placebo-controlled clinical trial. eClinicalMedicine 2021, 37, 100978.
  32. Gentile, P., The new regenerative and innovative strategies in hair loss. eClinicalMedicine 2021, 37, 100995.
  33. Bayart, C. B.; DeNiro, K. L.; Brichta, L.; Craiglow, B. G.; Sidbury, R., Topical Janus kinase inhibitors for the treatment of pediatric alopecia areata. J Am Acad Dermatol 2017, 77(1), 167-170.
  34. Jiménez-Acosta, F.; Ponce-Rodríguez, I., Follicular Unit Extraction for Hair Transplantation: An Update. Actas Dermosifiliogr 2017, 108(6), 532-537.
  35. Sung, J.-H., Effective and economical cell therapy for hair regeneration. Biomed Pharmacother 2023, 157, 113988.
  36. Correia, M.; Lopes, J.; Lopes, D.; Melero, A.; Makvandi, P.; Veiga, F.; Coelho, J. F. J.; Fonseca, A. C.; Paiva-Santos, A. C., Nanotechnology-based techniques for hair follicle regeneration. Biomaterials 2023, 302, 122348.
  37. Ali, A.; Phull, A.-R.; Zia, M., Elemental zinc to zinc nanoparticles: is ZnO NPs crucial for life? Synthesis, toxicological, and environmental concerns. Nano Rev 2018, 7(5), 413-441.
  38. Feldman, P. R.; Fiebig, K. M.; Piwko, C.; Mints, B. M.; Brown, D.; Cahan, D. J.; Guevara-Aguirre, J., Safety and efficacy of ALRV5XR in men with androgenetic alopecia: A randomised, double-blinded, placebo-controlled clinical trial. eClinicalMedicine 2021, 40, 101124.
  39. Stefanis, A. J.; Groh, T.; Arenbergerova, M.; Arenberger, P.; Bauer, P. O., Stromal Vascular Fraction and its Role in the Management of Alopecia: A Review. J Clin Aesthet Dermatol 2019, 12(11), 35-44.
  40. Shaikh, Z. S. A.; Patel, B. A. A.; Patil, S. G.; Maniyar, A. R. S., Nanotechnology-Based Strategies for Hair Follicle Regeneration in Androgenetic Alopecia. Mater Proc 2023, 14(1), 57.
  41. Gupta, A. K.; Wang, T.; Rapaport, J. A., Systematic review of exosome treatment in hair restoration: Preliminary evidence, safety, and future directions. J Cosmet Dermatol 2023, 22(9), 2424-2433.
  42. Zhou, Y.; Jia, L.; Zhou, D.; Chen, G.; Fu, Q.; Li, N., Advances in microneedles research based on promoting hair regrowth. J Control Release 2023, 353, 965-974.
  43. Novina, C. D.; Sharp, P. A., The RNAi revolution. Nature 2004, 430(6996), 161-164.
  44. Zhu, Y.; Zhu, L.; Wang, X.; Jin, H., RNA-based therapeutics: an overview and prospectus. Cell Death Dis 2022, 13(7), 644.
  45. Lam, J. K.; Chow, M. Y.; Zhang, Y.; Leung, S. W., siRNA Versus miRNA as Therapeutics for Gene Silencing. Mol Ther Nucleic Acids 2015, 4(9), e252.
  46. Rao, D. D.; Vorhies, J. S.; Senzer, N.; Nemunaitis, J., siRNA vs. shRNA: Similarities and differences. Adv Drug Deliv Rev 2009, 61(9), 746-759.
  47. Akinc, A.; Maier, M. A.; Manoharan, M.; Fitzgerald, K.; Jayaraman, M.; Barros, S.; Ansell, S.; Du, X.; Hope, M. J.; Madden, T. D.; Mui, B. L.; Semple, S. C.; Tam, Y. K.; Ciufolini, M.; Witzigmann, D.; Kulkarni, J. A.; van der Meel, R.; Cullis, P. R., The Onpattro story and the clinical translation of nanomedicines containing nucleic acid-based drugs. Nat Nanotechnol 2019, 14(12), 1084-1087.
  48. Syed, Y. Y., Givosiran: A Review in Acute Hepatic Porphyria. Drugs 2021, 81(7), 841-848.
  49. Scott, L. J.; Keam, S. J., Lumasiran: First Approval. Drugs 2021, 81(2), 277-282.
  50. Keam, S. J., Vutrisiran: First Approval. Drugs 2022, 82(13), 1419-1425.
  51. Migliorati, J. M.; Jin, J.; Zhong, X.-b., siRNA drug Leqvio (inclisiran) to lower cholesterol. Trends Pharmacol Sci 2022, 43(5), 455-456.
  52. Zhang, L.; Liang, Y.; Liang, G.; Tian, Z.; Zhang, Y.; Liu, Z.; Ji, X., The therapeutic prospects of N-acetylgalactosamine-siRNA conjugates. Front Pharmacol 2022, 13, 1090237.
  53. Traber, G. M.; Yu, A.-M., RNAi-Based Therapeutics and Novel RNA Bioengineering Technologies. J Pharmacol Exp Ther 2023, 384(1), 133-154.
  54. Maraganore, J., Reflections on Alnylam. Nat Biotechnol 2022, 40(5), 641-650.
  55. Ohrt, T.; Merkle, D.; Birkenfeld, K.; Echeverri, C. J.; Schwille, P., In situ fluorescence analysis demonstrates active siRNA exclusion from the nucleus by Exportin 5. Nucleic Acids Res 2006, 34(5), 1369-1380.
  56. Pi, F.; Binzel, D. W.; Lee, T. J.; Li, Z.; Sun, M.; Rychahou, P.; Li, H.; Haque, F.; Wang, S.; Croce, C. M.; Guo, B.; Evers, B. M.; Guo, P., Nanoparticle orientation to control RNA loading and ligand display on extracellular vesicles for cancer regression. Nat Nanotechnol 2018, 13(1), 82-89.
  57. de Brito, E. C. D.; Frederico, A. B. T.; Azamor, T.; Melgaço, J. G.; da Costa Neves, P. C.; Bom, A.; Tilli, T. M.; Missailidis, S., Biotechnological Evolution of siRNA Molecules: From Bench Tool to the Refined Drug. Pharmaceuticals (Basel) 2022, 15(5), 575.
  58. Hu, X.-M.; Li, Z.-X.; Zhang, D.-Y.; Yang, Y.-C.; Fu, S.-a.; Zhang, Z.-Q.; Yang, R.-H.; Xiong, K., A systematic summary of survival and death signalling during the life of hair follicle stem cells. Stem Cell Res Ther 2021, 12(1), 453.
  59. Li, K. N.; Tumbar, T., Hair follicle stem cells as a skin-organizing signaling center during adult homeostasis. EMBO J 2021, 40(11), e107135.
  60. Paus, R.; Ito, N.; Takigawa, M.; Ito, T., The Hair Follicle and Immune Privilege. J Investig Dermatol Symp Proc 2003, 8(2), 188-194.
  61. Botchkareva, N. V.; Ahluwalia, G.; Shander, D., Apoptosis in the Hair Follicle. J Invest Dermatol 2006, 126(2), 258-264.
  62. Stenn, K. S.; Paus, R., Controls of Hair Follicle Cycling. Physiol Rev 2001, 81(1), 449-494.
  63. Csuka, D. A.; Csuka, E. A.; Juhász, M. L. W.; Sharma, A. N.; Mesinkovska, N. A., A systematic review on the lipid composition of human hair. Int J Dermatol 2023, 62(3), 404-415.
  64. Zheng, M.; Oh, S. H.; Choi, N.; Choi, Y. J.; Kim, J.; Sung, J.-H., CXCL12 inhibits hair growth through CXCR4. Biomed Pharmacother 2022, 150, 112996.
  65. Nakamura, M.; Jo, J.-i.; Tabata, Y.; Ishikawa, O., Controlled Delivery of T-box21 Small Interfering RNA Ameliorates Autoimmune Alopecia (Alopecia Areata) in a C3H/HeJ Mouse Model. Am J Pathol 2008, 172(3), 650-658.
  66. Ryu, Y. C.; Park, J.; Kim, Y.-R.; Choi, S.; Kim, G.-U.; Kim, E.; Hwang, Y.; Kim, H.; Han, G.; Lee, S.-H.; Choi, K.-Y., CXXC5 Mediates DHT-Induced Androgenetic Alopecia via PGD2. Cells 2023, 12(4), 555.
  67. Liu, Q.; Tang, Y.; Huang, Y.; Wang, J. a.; Yang, K.; Zhang, Y.; Pu, W.; Liu, J.; Shi, X.; Ma, Y.; Ni, C.; Zhang, Y.; Zhu, Y.; Li, H.; Wang, J.; Lin, J.; Wu, W., Insights into male androgenetic alopecia using comparative transcriptome profiling: hypoxia-inducible factor-1 and Wnt/β-catenin signalling pathways. Br J Dermatol 2022, 187(6), 936-947.
  68. Zhao, J.; Lin, H.; Wang, L.; Guo, K.; Jing, R.; Li, X.; Chen, Y.; Hu, Z.; Gao, S.; Xu, N., Suppression of FGF5 and FGF18 Expression by Cholesterol-Modified siRNAs Promotes Hair Growth in Mice. Front Pharmacol 2021, 12, 666860.
  69. Chow, L. S.; Gerszten, R. E.; Taylor, J. M.; Pedersen, B. K.; van Praag, H.; Trappe, S.; Febbraio, M. A.; Galis, Z. S.; Gao, Y.; Haus, J. M.; Lanza, I. R.; Lavie, C. J.; Lee, C.-H.; Lucia, A.; Moro, C.; Pandey, A.; Robbins, J. M.; Stanford, K. I.; Thackray, A. E.; Villeda, S.; Watt, M. J.; Xia, A.; Zierath, J. R.; Goodpaster, B. H.; Snyder, M. P., Exerkines in health, resilience and disease. Nat Rev Endocrinol 2022, 18(5), 273-289.
  70. Moon, I. J.; Yoon, H. K.; Kim, D.; Choi, M. E.; Han, S. H.; Park, J. H.; Hong, S. W.; Cho, H.; Lee, D.-K.; Won, C. H., Efficacy of Asymmetric siRNA Targeting Androgen Receptors for the Treatment of Androgenetic Alopecia. Mol Pharm 2023, 20(1), 128-135.
  71. Lin, G.; Yin, G.; Ye, J.; Pan, X.; Zhu, J.; Lin, B., RNA sequence analysis of dermal papilla cells’ regeneration in 3D culture. J Cell Mol Med 2020, 24(22), 13421-13430.
  72. Xiong, J.; Chen, G.; Liu, Z.; Wu, X.; Xu, S.; Xiong, J.; Ji, S.; Wu, M., Construction of regulatory network for alopecia areata progression and identification of immune monitoring genes based on multiple machine-learning algorithms. Precis Clin Med 2023, 6(2), pbad009.
  73. Uchida, Y.; Gherardini, J.; Pappelbaum, K.; Chéret, J.; Schulte-Mecklenbeck, A.; Gross, C. C.; Strbo, N.; Gilhar, A.; Rossi, A.; Funk, W.; Kanekura, T.; Almeida, L.; Bertolini, M.; Paus, R., Resident human dermal γδT-cells operate as stress-sentinels: Lessons from the hair follicle. J Autoimmun 2021, 124, 102711.
  74. Martinez-Lopez, A.; Montero-Vilchez, T.; Sierra-Sánchez, Á.; Molina-Leyva, A.; Arias-Santiago, S., Advanced Medical Therapies in the Management of Non-Scarring Alopecia: Areata and Androgenic Alopecia. Int J Mol Sci 2020, 21(21), 8390.
  75. Xiong, X.; Tu, S.; Wang, J.; Luo, S.; Yan, X., CXXC5: A novel regulator and coordinator of TGF-β, BMP and Wnt signaling. J Cell Mol Med 2019, 23(2), 740-749.
  76. Sadgrove, N.; Batra, S.; Barreto, D.; Rapaport, J., An Updated Etiology of Hair Loss and the New Cosmeceutical Paradigm in Therapy: Clearing ‘the Big Eight Strikes&rsquo. Cosmetics 2023, 10(4), 106.
  77. Burg, D.; Yamamoto, M.; Namekata, M.; Haklani, J.; Koike, K.; Halasz, M., Promotion of anagen, increased hair density and reduction of hair fall in a clinical setting following identification of FGF5-inhibiting compounds via a novel 2-stage process. Clinical, Clin Cosmet Investig Dermatol 2017, 10, 71-85.
  78. Guo, K.; Wang, L.; Zhong, Y.; Gao, S.; Jing, R.; Ye, J.; Zhang, K.; Fu, M.; Hu, Z.; Zhao, W.; Xu, N., Cucurbitacin promotes hair growth in mice by inhibiting the expression of fibroblast growth factor 18. Ann Transl Med 2022, 10(20), 1104.
  79. Leishman, E.; Howard, J. M.; Garcia, G. E.; Miao, Q.; Ku, A. T.; Dekker, J. D.; Tucker, H.; Nguyen, H., Foxp1 maintains hair follicle stem cell quiescence through regulation of Fgf18. Development 2013, 140(18), 3809-3818.
  80. Yun, S.-I.; Lee, S.-K.; Goh, E.-A.; Kwon, O. S.; Choi, W.; Kim, J.; Lee, M. S.; Choi, S. J.; Lim, S. S.; Moon, T. K.; Kim, S. H.; Kyong, K.; Nam, G.; Park, H.-O., Weekly treatment with SAMiRNA targeting the androgen receptor ameliorates androgenetic alopecia. Sci Rep 2022, 12(1), 1607.
  81. Friedrich, M.; Aigner, A., Therapeutic siRNA: State-of-the-Art and Future Perspectives. BioDrugs 2022, 36(5), 549-571.
  82. Naito, Y.; Ui-Tei, K., siRNA Design Software for a Target Gene-Specific RNA Interference. Front Genet 2012, 3, 102.
  83. Pushparaj, P. N.; Aarthi, J. J.; Manikandan, J.; Kumar, S. D., siRNA, miRNA, and shRNA: in vivo Applications. J Dent Res 2008, 87(11), 992-1003.
  84. Degors, I. M. S.; Wang, C.; Rehman, Z. U.; Zuhorn, I. S., Carriers Break Barriers in Drug Delivery: Endocytosis and Endosomal Escape of Gene Delivery Vectors. Acc Chem Res 2019, 52(7), 1750-1760.
  85. Qiu, C.; Xia, F.; Zhang, J.; Shi, Q.; Meng, Y.; Wang, C.; Pang, H.; Gu, L.; Xu, C.; Guo, Q.; Wang, J., Advanced Strategies for Overcoming Endosomal/Lysosomal Barrier in Nanodrug Delivery. Research 2023, 6, 0148.
  86. Kanasty, R. L.; Whitehead, K. A.; Vegas, A. J.; Anderson, D. G., Action and Reaction: The Biological Response to siRNA and Its Delivery Vehicles. Mol Ther 2012, 20(3), 513-524.
  87. Kaushal, A., Innate immune regulations and various siRNA modalities. Drug Deliv Transl Res 2023, 13(11), 2704-2718.
  88. Judge, A. D.; Sood, V.; Shaw, J. R.; Fang, D.; McClintock, K.; MacLachlan, I., Sequence-dependent stimulation of the mammalian innate immune response by synthetic siRNA. Nat Biotechnol 2005, 23(4), 457-462.
  89. Kaushal, A., Innate immune regulations and various siRNA modalities. Drug Deliv Transl Res 2023, 13(11), 2704-2718.
  90. Ran, F. A.; Hsu, P. D.; Wright, J.; Agarwala, V.; Scott, D. A.; Zhang, F., Genome engineering using the CRISPR-Cas9 system. Nat Protoc 2013, 8(11), 2281-2308.
  91. Chen, P. Y.; Weinmann, L.; Gaidatzis, D.; Pei, Y.; Zavolan, M.; Tuschl, T.; Meister, G., Strand-specific 5'-O-methylation of siRNA duplexes controls guide strand selection and targeting specificity. RNA 2008, 14(2), 263-74.
  92. Ozcan, G.; Ozpolat, B.; Coleman, R. L.; Sood, A. K.; Lopez-Berestein, G., Preclinical and clinical development of siRNA-based therapeutics. Adv Drug Deliv Rev 2015, 87, 108-119.
  93. Ablon, G., A 6-month, randomized, double-blind, placebo-controlled study evaluating the ability of a marine complex supplement to promote hair growth in men with thinning hair. J Cosmet Dermatol 2016, 15(4), 358-366.
  94. Devjani, S.; Ezemma, O.; Kelley, K. J.; Stratton, E.; Senna, M., Androgenetic Alopecia: Therapy Update. Drugs 2023, 83(8), 701-715.
  95. Ranjbar, S.; Zhong, X.-b.; Manautou, J.; Lu, X., A holistic analysis of the intrinsic and delivery-mediated toxicity of siRNA therapeutics. Adv Drug Deliv Rev 2023, 201, 115052.
  96. Parmar., I. S. P. A. U. M. M., Small Interfering RNA (siRNA) Therapy [Updated 2023 Jun 3]. Treasure Island (FL): StatPearls Publishing: 2023; Vol. StatPearls [Internet].
  97. Moon, I. J.; Yoon, H. K.; Kim, D.; Choi, M. E.; Han, S. H.; Park, J. H.; Hong, S. W.; Cho, H.; Lee, D. K.; Won, C. H., Efficacy of Asymmetric siRNA Targeting Androgen Receptors for the Treatment of Androgenetic Alopecia. Mol Pharm 2023, 20(1), 128-135.
  98. FDA, Drug Products, Including Biological Products, that Contain Nanomaterials. 2022.
  99. FDA, Considering Whether an FDA-Regulated Product Involves the Application of Nanotechnology. 2014.
  100. Boettcher, M.; McManus, M. T., Choosing the Right Tool for the Job: RNAi, TALEN, or CRISPR. Mol Cell 2015, 58(4), 575-85.
  101. Peretz, L.; Besser, E.; Hajbi, R.; Casden, N.; Ziv, D.; Kronenberg, N.; Gigi, L. B.; Sweetat, S.; Khawaled, S.; Aqeilan, R.; Behar, O., Combined shRNA over CRISPR/cas9 as a methodology to detect off-target effects and a potential compensatory mechanism. Sci Rep 2018, 8(1), 93.
  102. Unniyampurath, U.; Pilankatta, R.; Krishnan, M. N., RNA Interference in the Age of CRISPR: Will CRISPR Interfere with RNAi? Int J Mol Sci 2016, 17(3), 291.
  103. Carroll, D., RNA in Therapeutics: CRISPR in the Clinic. Mol Cells 2023, 46(1), 4-9.
  104. Kingwell, K., First CRISPR therapy seeks landmark approval. Nat Rev Drug Discov 2023, 22(5), 339-341.
  105. Ledford, H., Is CRISPR safe? Genome editing gets its first FDA scrutiny. Nature 2023, 623(7986), 234-235.
  106. Beg, S.; Rahman, M.; Kohli, K., Quality-by-design approach as a systematic tool for the development of nanopharmaceutical products. Drug Discov Today 2019, 24(3), 717-725.
  107. Chernikov, I. V.; Vlassov, V. V.; Chernolovskaya, E. L., Current Development of siRNA Bioconjugates: From Research to the Clinic. Front Pharmacol 2019, 10, 444.
  108. Niazi, S. K., RNA Therapeutics: A Healthcare Paradigm Shift. Biomedicines 2023, 11(5), 1275.
  109. Ahn, I.; Kang, C. S.; Han, J., Where should siRNAs go: applicable organs for siRNA drugs. Exp Mol Med 2023, 55(7), 1283-1292.
  110. Ali Zaidi, S. S.; Fatima, F.; Ali Zaidi, S. A.; Zhou, D.; Deng, W.; Liu, S., Engineering siRNA therapeutics: challenges and strategies. J Nanobiotechnology 2023, 21(1), 381.
  111. Huang, Y.; Cheng, Q.; Ji, J. L.; Zheng, S.; Du, L.; Meng, L.; Wu, Y.; Zhao, D.; Wang, X.; Lai, L.; Cao, H.; Xiao, K.; Gao, S.; Liang, Z., Pharmacokinetic Behaviors of Intravenously Administered siRNA in Glandular Tissues. Theranostics 2016, 6(10), 1528-41.
  112. Ranasinghe, P.; Addison, M. L.; Dear, J. W.; Webb, D. J., Small interfering RNA: Discovery, pharmacology and clinical development—An introductory review. Br J Pharmacol 2023, 180(21), 2697-2720.
  113. Marschall, A. L. J., Targeting the Inside of Cells with Biologicals: Chemicals as a Delivery Strategy. BioDrugs 2021, 35(6), 643-671.
  114. Ren, K.; Liu, Y.; Wu, J.; Zhang, Y.; Zhu, J.; Yang, M.; Ju, H., A DNA dual lock-and-key strategy for cell-subtype-specific siRNA delivery. Nat Commun 2016, 7(1), 13580.
  115. Hedlund, H.; Du Rietz, H.; Johansson, J. M.; Eriksson, H. C.; Zedan, W.; Huang, L.; Wallin, J.; Wittrup, A., Single-cell quantification and dose-response of cytosolic siRNA delivery. Nat Commun 2023, 14(1), 1075.
  116. Teo, S. L. Y.; Rennick, J. J.; Yuen, D.; Al-Wassiti, H.; Johnston, A. P. R.; Pouton, C. W., Unravelling cytosolic delivery of cell penetrating peptides with a quantitative endosomal escape assay. Nat Commun 2021, 12(1), 3721.
  117. Jing, X.; Arya, V.; Reynolds, K. S.; Rogers, H., Clinical Pharmacology of RNAi-based Therapeutics: A Summary Based On FDA-Approved Small-interfering RNAs. Drug Metab Dispos 2022, DMD-MR-2022-001107.
  118. Meng, Z.; Lu, M., RNA Interference-Induced Innate Immunity, Off-Target Effect, or Immune Adjuvant? Front Immunol 2017, 8, 331.
  119. Bartoszewski, R.; Sikorski, A. F., Editorial focus: understanding off-target effects as the key to successful RNAi therapy. Cell Mol Biol Lett 2019, 24(1), 69.
  120. Lin, X.; Ruan, X.; Anderson, M. G.; McDowell, J. A.; Kroeger, P. E.; Fesik, S. W.; Shen, Y., siRNA-mediated off-target gene silencing triggered by a 7 nt complementation. Nucleic Acids Res 2005, 33(14), 4527-4535.
  121. Suter, S. R.; Sheu-Gruttadauria, J.; Schirle, N. T.; Valenzuela, R.; Ball-Jones, A. A.; Onizuka, K.; MacRae, I. J.; Beal, P. A., Structure-Guided Control of siRNA Off-Target Effects. J Am Chem Soc 2016, 138(28), 8667-8669.
  122. Dominska, M.; Dykxhoorn, D. M., Breaking down the barriers: siRNA delivery and endosome escape. J Cell Sci 2010, 123(8), 1183-1189.
  123. Dowdy, S. F., Endosomal escape of RNA therapeutics: How do we solve this rate-limiting problem? RNA 2023, 29(4), 396-401.
  124. Bruno, K., Using drug-excipient interactions for siRNA delivery. Adv Drug Deliv Rev 2011, 63(13), 1210-1226.
  125. Dong, Y.; Siegwart, D. J.; Anderson, D. G., Strategies, design, and chemistry in siRNA delivery systems. Adv Drug Deliv Rev 2019, 144, 133-147.
  126. Mendonça, M. C. P.; Kont, A.; Kowalski, P. S.; O'Driscoll, C. M., Design of lipid-based nanoparticles for delivery of therapeutic nucleic acids. Drug Discov Today 2023, 28(3), 103505.
  127. Dong, Y.; Love, K. T.; Dorkin, J. R.; Sirirungruang, S.; Zhang, Y.; Chen, D.; Bogorad, R. L.; Yin, H.; Chen, Y.; Vegas, A. J.; Alabi, C. A.; Sahay, G.; Olejnik, K. T.; Wang, W.; Schroeder, A.; Lytton-Jean, A. K. R.; Siegwart, D. J.; Akinc, A.; Barnes, C.; Barros, S. A.; Carioto, M.; Fitzgerald, K.; Hettinger, J.; Kumar, V.; Novobrantseva, T. I.; Qin, J.; Querbes, W.; Koteliansky, V.; Langer, R.; Anderson, D. G., Lipopeptide nanoparticles for potent and selective siRNA delivery in rodents and nonhuman primates. Proc Natl Acad Sci USA 2014, 111(11), 3955-3960.
  128. Love, K. T.; Mahon, K. P.; Levins, C. G.; Whitehead, K. A.; Querbes, W.; Dorkin, J. R.; Qin, J.; Cantley, W.; Qin, L. L.; Racie, T.; Frank-Kamenetsky, M.; Yip, K. N.; Alvarez, R.; Sah, D. W. Y.; de Fougerolles, A.; Fitzgerald, K.; Koteliansky, V.; Akinc, A.; Langer, R.; Anderson, D. G., Lipid-like materials for low-dose, in vivo gene silencing. Proc Natl Acad Sci USA 2010, 107(5), 1864-1869.
  129. Zhang, X.; Zhao, W.; Nguyen, G. N.; Zhang, C.; Zeng, C.; Yan, J.; Du, S.; Hou, X.; Li, W.; Jiang, J.; Deng, B.; McComb, D. W.; Dorkin, R.; Shah, A.; Barrera, L.; Gregoire, F.; Singh, M.; Chen, D.; Sabatino, D. E.; Dong, Y., Functionalized lipid-like nanoparticles for in vivo mRNA delivery and base editing. Sci Adv 2020, 6(34), eabc2315.
  130. Yonezawa, S.; Koide, H.; Asai, T., Recent advances in siRNA delivery mediated by lipid-based nanoparticles. Adv Drug Deliv Rev 2020, 154-155, 64-78.
  131. Pereira, M. N.; Ushirobira, C. Y.; Cunha-Filho, M. S.; Gelfuso, G. M.; Gratieri, T., Nanotechnology advances for hair loss. Ther Deliv 2018, 9(8), 593-603.
  132. Castro, A. R.; Portinha, C.; Logarinho, E., The booming business of hair loss. Trends Biotechnol 2023, 41(6), 731-735.
  133. Pereira-Silva, M.; Martins, A. M.; Sousa-Oliveira, I.; Ribeiro, H. M.; Veiga, F.; Marto, J.; Paiva-Santos, A. C., Nanomaterials in hair care and treatment. Acta Biomater 2022, 142, 14-35.
  134. Mendes, B. B.; Conniot, J.; Avital, A.; Yao, D.; Jiang, X.; Zhou, X.; Sharf-Pauker, N.; Xiao, Y.; Adir, O.; Liang, H.; Shi, J.; Schroeder, A.; Conde, J., Nanodelivery of nucleic acids. Nat Rev Methods Primers 2022, 2(1), 24.
  135. Buffoli, B.; Rinaldi, F.; Labanca, M.; Sorbellini, E.; Trink, A.; Guanziroli, E.; Rezzani, R.; Rodella, L. F., The human hair: from anatomy to physiology. Int J Dermatol 2014, 53(3), 331-341.
  136. Jeyaraman, M.; Muthu, S.; Sharma, S.; Ganta, C.; Ranjan, R.; Jha, S. K., Nanofat: A therapeutic paradigm in regenerative medicine. World J Stem Cells 2021, 13(11), 1733-1746.
  137. Castillo Cruz, B.; Flores Colón, M.; Rabelo Fernandez, R. J.; Vivas-Mejia, P. E.; Barletta, G. L., A Fresh Look at the Potential of Cyclodextrins for Improving the Delivery of siRNA Encapsulated in Liposome Nanocarriers. ACS Omega 2022, 7(4), 3731-3737.
  138. Konrádsdóttir, F.; Ogmundsdóttir, H.; Sigurdsson, V.; Loftsson, T., Drug targeting to the hair follicles: a cyclodextrin-based drug delivery. AAPS PharmSciTech 2009, 10(1), 266-9.
  139. Zhang, Z.; Li, W.; Chang, D.; Wei, Z.; Wang, E.; Yu, J.; Xu, Y.; Que, Y.; Chen, Y.; Fan, C.; Ma, B.; Zhou, Y.; Huan, Z.; Yang, C.; Guo, F.; Chang, J., A combination therapy for androgenic alopecia based on quercetin and zinc/copper dual-doped mesoporous silica nanocomposite microneedle patch. Bioact Mater 2023, 24, 81-95.
  140. English, R. S.; Ruiz, S.; DoAmaral, P., Microneedling and Its Use in Hair Loss Disorders: A Systematic Review. Dermatol Ther 2022, 12(1), 41-60.
  141. Kanasty, R.; Dorkin, J. R.; Vegas, A.; Anderson, D., Delivery materials for siRNA therapeutics. Nat Mater 2013, 12(11), 967-977.
  142. Stenn, K. S.; Karnik, P., Lipids to the top of hair biology. J Invest Dermatol 2010, 130(5), 1205-1207.
  143. Shimomura, Y.; Christiano, A. M., Biology and Genetics of Hair. Annu Rev Genomics Hum Genet 2010, 11(1), 109-132.
  144. Aikawa, S.; Hashimoto, T.; Kano, K.; Aoki, J., Lysophosphatidic acid as a lipid mediator with multiple biological actions. J Biochem 2014, 157(2), 81-89.
  145. Schmidt, B.; Horsley, V., Unravelling hair follicle–adipocyte communication. Exp Dermatol 2012, 21(11), 827-830.
  146. Shah, S.; Dhawan, V.; Holm, R.; Nagarsenker, M. S.; Perrie, Y., Liposomes: Advancements and innovation in the manufacturing process. Adv Drug Deliv Rev 2020, 154-155, 102-122.
  147. Abu-Huwaij, R.; Zidan, A. N., Unlocking the potential of cosmetic dermal delivery with ethosomes: A comprehensive review. J Cosmet Dermatol 2024, 23(1), 17-26.
  148. Chauhan, N.; Vasava, P.; Khan, S. L.; Siddiqui, F. A.; Islam, F.; Chopra, H.; Emran, T. B., Ethosomes: A novel drug carrier. Ann Med Surg 2022, 82, 104595.
  149. Mawazi, S. M.; Ann, T. J.; Widodo, R. T., Application of Niosomes in Cosmetics: A Systematic Review. Cosmetics 2022, 9(6), 127.
  150. Xia, Y.; Tian, J.; Chen, X., Effect of surface properties on liposomal siRNA delivery. Biomaterials 2016, 79, 56-68.
  151. Li, L.; Hoffman, R. M., The feasibility of targeted selective gene therapy of the hair follicle. Nat Medicine 1995, 1(7), 705-706.
  152. Madhunithya, E.; Venkatesh, G.; Shyamala, G.; Manjari, V.; Ramesh, S.; Karuppaiah, A.; Sankar, V., Development of ethosome comprising combined herbal extracts and its effect on hair growth. Adv Tradit Med (ADTM) 2021, 21(1), 131-141.
  153. Paiva-Santos, A. C.; Silva, A. L.; Guerra, C.; Peixoto, D.; Pereira-Silva, M.; Zeinali, M.; Mascarenhas-Melo, F.; Castro, R.; Veiga, F., Ethosomes as Nanocarriers for the Development of Skin Delivery Formulations. Pharm Res 2021, 38(6), 947-970.
  154. G, D. B.; P, V. L., Recent advances of non-ionic surfactant-based nano-vesicles (niosomes and proniosomes): a brief review of these in enhancing transdermal delivery of drug. Future J Pharm Sci 2020, 6(1), 100.
  155. Momekova, D. B.; Gugleva, V. E.; Petrov, P. D., Nanoarchitectonics of Multifunctional Niosomes for Advanced Drug Delivery. ACS Omega 2021, 6(49), 33265-33273.
  156. Teeranachaideekul, V.; Parichatikanond, W.; Junyaprasert, V. B.; Morakul, B., Pumpkin Seed Oil-Loaded Niosomes for Topical Application: 5α-Reductase Inhibitory, Anti-Inflammatory, and In Vivo Anti-Hair Loss Effects. Pharmaceuticals 2022, 15(8), 930.
  157. Chen, S.; Hanning, S.; Falconer, J.; Locke, M.; Wen, J., Recent advances in non-ionic surfactant vesicles (niosomes): Fabrication, characterization, pharmaceutical and cosmetic applications. Eur J Pharm Biopharm 2019, 144, 18-39.
  158. Khan, R.; Irchhaiya, R., Niosomes: a potential tool for novel drug delivery. J Pharm Investig 2016, 46(3), 195-204.
  159. O’Brien, K.; Breyne, K.; Ughetto, S.; Laurent, L. C.; Breakefield, X. O., RNA delivery by extracellular vesicles in mammalian cells and its applications. Nat Rev Mol Cell Biol 2020, 21(10), 585-606.
  160. Tenchov, R.; Sasso, J. M.; Wang, X.; Liaw, W.-S.; Chen, C.-A.; Zhou, Q. A., Exosomes─Nature’s Lipid Nanoparticles, a Rising Star in Drug Delivery and Diagnostics. ACS Nano 2022, 16(11), 17802-17846.
  161. Herrmann, I. K.; Wood, M. J. A.; Fuhrmann, G., Extracellular vesicles as a next-generation drug delivery platform. Nat Nanotechnol 2021, 16(7), 748-759.
  162. Murphy, D. E.; de Jong, O. G.; Brouwer, M.; Wood, M. J.; Lavieu, G.; Schiffelers, R. M.; Vader, P., Extracellular vesicle-based therapeutics: natural versus engineered targeting and trafficking. Exp Mol Med 2019, 51(3), 1-12.
  163. Edgar, J. R., Q&A: What are exosomes, exactly? BMC Biology 2016, 14(1), 46.
  164. Zhang, B.; Gong, J.; He, L.; Khan, A.; Xiong, T.; Shen, H.; Li, Z., Exosomes based advancements for application in medical aesthetics. Front Bioeng Biotechnol 2022, 10, 1083640.
  165. Li, J.; Zhao, B.; Yao, S.; Dai, Y.; Zhang, X.; Yang, N.; Bao, Z.; Cai, J.; Chen, Y.; Wu, X., Dermal PapillaCell-Derived Exosomes Regulate Hair Follicle Stem Cell Proliferation via LEF1. Int J Mol Sci 2023, 24(4), 3961.
  166. Liang, Y.; Tang, X.; Zhang, X.; Cao, C.; Yu, M.; Wan, M., Adipose Mesenchymal Stromal Cell-Derived Exosomes Carrying MiR-122-5p Antagonize the Inhibitory Effect of Dihydrotestosterone on Hair Follicles by Targeting the TGF-β1/SMAD3 Signaling Pathway. Int J Mol Sci 2023, 24(6), 5703.
  167. Shimizu, Y.; Ntege, E. H.; Sunami, H.; Inoue, Y., Regenerative medicine strategies for hair growth and regeneration: A narrative review of literature. Regen Ther 2022, 21, 527-539.
  168. Yuan, A.-R.; Bian, Q.; Gao, J.-Q., Current advances in stem cell-based therapies for hair regeneration. Eur J Pharmacol 2020, 881, 173197.
  169. Gomes, M. J.; Martins, S.; Ferreira, D.; Segundo, M. A.; Reis, S., Lipid nanoparticles for topical and transdermal application for alopecia treatment: development, physicochemical characterization, and in vitro release and penetration studies. Int J Nanomedicine 2014, 9, 1231-42.
  170. Lauterbach, A.; Müller-Goymann, C. C., Applications and limitations of lipid nanoparticles in dermal and transdermal drug delivery via the follicular route. Eur J Pharm Biopharm 2015, 97, 152-163.
  171. Hatem, S.; Nasr, M.; Moftah, N. H.; Ragai, M. H.; Geneidi, A. S.; Elkheshen, S. A., Clinical cosmeceutical repurposing of melatonin in androgenic alopecia using nanostructured lipid carriers prepared with antioxidant oils. Expert Opin Drug Deliv 2018, 15(10), 927-935.
  172. Oliveira, P. M.; Alencar-Silva, T.; Pires, F. Q.; Cunha-Filho, M.; Gratieri, T.; Carvalho, J. L.; Gelfuso, G. M., Nanostructured lipid carriers loaded with an association of minoxidil and latanoprost for targeted topical therapy of alopecia. Eur J Pharm Biopharm 2022, 172, 78-88.
  173. Prasertpol, T.; Tiyaboonchai, W., Nanostructured lipid carriers: A novel hair protective product preventing hair damage and discoloration from UV radiation and thermal treatment. J Photochem Photobiol B 2020, 204, 111769.
  174. Pereira, M. N.; Tolentino, S.; Pires, F. Q.; Anjos, J. L. V.; Alonso, A.; Gratieri, T.; Cunha-Filho, M.; Gelfuso, G. M., Nanostructured lipid carriers for hair follicle-targeted delivery of clindamycin and rifampicin to hidradenitis suppurativa treatment. Colloids Surf B Biointerfaces 2021, 197, 111448.
  175. Sainaga Jyothi, V. G. S.; Ghouse, S. M.; Khatri, D. K.; Nanduri, S.; Singh, S. B.; Madan, J., Lipid nanoparticles in topical dermal drug delivery: Does chemistry of lipid persuade skin penetration? J Drug Deliv Sci Technol 2022, 69, 103176.
  176. Sala, M.; Diab, R.; Elaissari, A.; Fessi, H., Lipid nanocarriers as skin drug delivery systems: Properties, mechanisms of skin interactions and medical applications. Int J Pharm 2018, 535(1), 1-17.
  177. Salvi, V. R.; Pawar, P., Nanostructured lipid carriers (NLC) system: A novel drug targeting carrier. J Drug Deliv Sci Technol 2019, 51, 255-267.
  178. Khan, S.; Sharma, A.; Jain, V., An Overview of Nanostructured Lipid Carriers and its Application in Drug Delivery through Different Routes. Adv Pharm Bull 2023, 13(3), 446-460.
  179. Kumar, R.; Dkhar, D. S.; Kumari, R.; Divya; Mahapatra, S.; Dubey, V. K.; Chandra, P., Lipid based nanocarriers: Production techniques, concepts, and commercialization aspect. J Drug Deliv Sci Technol 2022, 74, 103526.
  180. Dubey, S. K.; Dey, A.; Singhvi, G.; Pandey, M. M.; Singh, V.; Kesharwani, P., Emerging trends of nanotechnology in advanced cosmetics. Colloids Surf B Biointerfaces 2022, 214, 112440.
  181. Ghitman, J.; Voicu, S. I., Controlled drug delivery mediated by cyclodextrin-based supramolecular self-assembled carriers: From design to clinical performances. Carbohydr Polym Technol Appl 2023, 5, 100266.
  182. Sufianov, A.; Beilerli, A.; Kudriashov, V.; Ilyasova, T.; Wenjie, B.; Beylerli, O., Advances in transdermal siRNAs delivery: A review of current research progress. Non-coding RNA Res 2023, 8(3), 392-400.
  183. Marcovici, G.; Bauman, A., An Uncontrolled Case Series Using a Botanically Derived, β-Cyclodextrin Inclusion Complex in Two Androgenetic Alopecia-Affected Male Subjects. Cosmetics 2020, 7(3), 65.
  184. Ferreira, L.; Mascarenhas-Melo, F.; Rabaça, S.; Mathur, A.; Sharma, A.; Giram, P. S.; Pawar, K. D.; Rahdar, A.; Raza, F.; Veiga, F.; Mazzola, P. G.; Paiva-Santos, A. C., Cyclodextrin-based dermatological formulations: Dermopharmaceutical and cosmetic applications. Colloids Surf B Biointerfaces 2023, 221, 113012.
  185. Muankaew, C.; Loftsson, T., Cyclodextrin-Based Formulations: A Non-Invasive Platform for Targeted Drug Delivery. Basic Clin Pharmacol Toxicol 2018, 122(1), 46-55.
  186. Wang, S.; Wei, Y.; Wang, Y.; Cheng, Y., Cyclodextrin regulated natural polysaccharide hydrogels for biomedical applications-a review. Carbohydr Polym 2023, 313, 120760.
  187. Morin-Crini, N.; Fourmentin, S.; Fenyvesi, É.; Lichtfouse, E.; Torri, G.; Fourmentin, M.; Crini, G., 130 years of cyclodextrin discovery for health, food, agriculture, and the industry: a review. Environ Chem Lett 2021, 19(3), 2581-2617.
  188. Davis, M. E., The First Targeted Delivery of siRNA in Humans via a Self-Assembling, Cyclodextrin Polymer-Based Nanoparticle: From Concept to Clinic. Mol Pharm 2009, 6(3), 659-668.
  189. Peng, H.; Cui, B.; Li, G.; Wang, Y.; Li, N.; Chang, Z.; Wang, Y., A multifunctional β-CD-modified Fe3O4@ZnO:Er3+,Yb3+ nanocarrier for antitumor drug delivery and microwave-triggered drug release. Mater Sci Eng C 2015, 46, 253-263.
  190. Yadwade, R.; Gharpure, S.; Ankamwar, B., Nanotechnology in cosmetics pros and cons. Nano Express 2021, 2(2), 022003.
  191. Mishra, P.; Handa, M.; Ujjwal, R. R.; Singh, V.; Kesharwani, P.; Shukla, R., Potential of nanoparticulate based delivery systems for effective management of alopecia. Colloids Surf B Biointerfaces 2021, 208, 112050.
  192. Kondrakhina, I. N.; Verbenko, D. A.; Zatevalov, A. M.; Gatiatulina, E. R.; Nikonorov, A. A.; Deryabin, D. G.; Kubanov, A. A., Plasma Zinc Levels in Males with Androgenetic Alopecia as Possible Predictors of the Subsequent Conservative Therapy’s Effectiveness. Diagnostics 2020, 10(5), 336.
  193. Zou, P.; Du, Y.; Yang, C.; Cao, Y., Trace element zinc and skin disorders. Front Med 2023, 9, 1093868.
  194. Draelos, Z. D.; Kenneally, D. C.; Hodges, L. T.; Billhimer, W.; Copas, M.; Margraf, C., A Comparison of Hair Quality and Cosmetic Acceptance Following the Use of Two Anti-Dandruff Shampoos. J Investig Dermatol Symp Proc 2005, 10(3), 201-204.
  195. Poojary, P. V.; Sarkar, S.; Poojary, A. A.; Mallya, P.; Selvaraj, R.; Koteshwara, A.; Aranjani, J. M.; Lewis, S., Novel anti-dandruff shampoo incorporated with ketoconazole-coated zinc oxide nanoparticles using green tea extract. J Cosmet Dermatol 2024, 23, 563-575.
  196. Trüeb, R. M.; Henry, J. P.; Davis, M. G.; Schwartz, J. R., Scalp Condition Impacts Hair Growth and Retention via Oxidative Stress. Int J Trichology 2018, 10(6), 262-270.
  197. Lin, Y.; Shao, R.; Xiao, T.; Sun, S., Promotion of Hair Regrowth by Transdermal Dissolvable Microneedles Loaded with Rapamycin and Epigallocatechin Gallate Nanoparticles. Pharmaceutics 2022, 14(7), 1404.
  198. Holbrook, K. A.; Odland, G. F., Regional Differences in the Thickness (Cell Layers) of the Human Stratum Corneum: An Ultrastructural Analysis. J Invest Dermatol 1974, 62(4), 415-422.
  199. Chen, W.; Li, H.; Shi, D.; Liu, Z.; Yuan, W., Microneedles As a Delivery System for Gene Therapy. Front Pharmacol 2016, 7, 137.
  200. Deng, Y.; Chen, J.; Zhao, Y.; Yan, X.; Zhang, L.; Choy, K.; Hu, J.; Sant, H. J.; Gale, B. K.; Tang, T., Transdermal Delivery of siRNA through Microneedle Array. Sci Rep 2016, 6(1), 21422.
  201. Xiang, H.; Xu, S.; Zhang, W.; Xue, X.; Li, Y.; Lv, Y.; Chen, J.; Miao, X., Dissolving microneedles for alopecia treatment. Colloids Surf B Biointerfaces 2023, 229, 113475.
  202. Zhao, Y.; Tian, Y.; Ye, W.; Wang, X.; Huai, Y.; Huang, Q.; Chu, X.; Deng, X.; Qian, A., A lipid–polymer hybrid nanoparticle (LPN)-loaded dissolving microneedle patch for promoting hair regrowth by transdermal miR-218 delivery. Biomater Sci 2023, 11(1), 140-152.
  203. Kozielski, K. L.; Tzeng, S. Y.; Green, J. J., Bioengineered nanoparticles for siRNA delivery. Wiley Interdiscip Rev Nanomed Nanobiotechnol 2013, 5(5), 449-468.
  204. Patzelt, A.; Lademann, J., Drug delivery to hair follicles. Expert Opin Drug Deliv 2013, 10(6), 787-797.
  205. Williford, J.-M.; Wu, J.; Ren, Y.; Archang, M. M.; Leong, K. W.; Mao, H.-Q., Recent Advances in Nanoparticle-Mediated siRNA Delivery. Annu Rev Biomed Eng 2014, 16(1), 347-370.
  206. Childs-Disney, J. L.; Yang, X.; Gibaut, Q. M. R.; Tong, Y.; Batey, R. T.; Disney, M. D., Targeting RNA structures with small molecules. Nat Rev Drug Discov 2022, 21(10), 736-762.
  207. Salmikangas, P., Design and optimisation of a quality target product pro le for ATMPs. Regulatory Rapporteur 2019, 16(2), 4-7.
  208. Wang, J.; Fu, Y.; Huang, W.; Biswas, R.; Banerjee, A.; Broussard, J. A.; Zhao, Z.; Wang, D.; Bjerke, G.; Raghavan, S.; Yan, J.; Green, K. J.; Yi, R., MicroRNA-205 promotes hair regeneration by modulating mechanical properties of hair follicle stem cells. Proc Natl Acad Sci USA 2023, 120(22), e2220635120.

Sincerely,

Su-Eon Jin, Ph.D.

Advisory member, Epi Biotech Co., Ltd.

Jong-Hyuk Sung, Ph.D.

CEO, Epi Biotech Co., Ltd.

Round 2

Reviewer 1 Report

Comments and Suggestions for Authors

The authors have addressed my concerns.